# REGMIX: DATA MIXTURE AS REGRESSION FOR LANGUAGE MODEL PRE-TRAINING

**Qian Liu**[1*]  **Xiaosen Zheng**[2*]  **Niklas Muennighoff**[3,4]  **Guangtao Zeng**[5]
**Longxu Dou**[1]  **Tianyu Pang**[1]  **Jing Jiang**[2]  **Min Lin**[1]
[1]Sea AI Lab  [2]SMU  [3]Contextual AI  [4]Stanford University  [5]SUTD
liuqian.sea@gmail.com; xszheng.2020@phdcs.smu.edu.sg

## ABSTRACT

The data mixture for large language model pre-training significantly impacts performance, yet how to determine an effective mixture remains unclear. We propose REGMIX to automatically identify a high-performing data mixture by formulating it as a regression task. REGMIX trains many small models on diverse data mixtures, uses regression to predict performance of unseen mixtures, and applies the best predicted mixture to train a large-scale model with orders of magnitude more compute. To empirically validate REGMIX, we train 512 models with 1M parameters for 1B tokens to fit the regression model and predict the best data mixture. Using this mixture we train a 1B parameter model for 25B tokens (i.e. $1000\times$ larger and $25\times$ longer) which we find performs best among 64 candidate 1B parameter models with other mixtures. Furthermore, REGMIX consistently outperforms human selection in experiments involving models up to 7B models trained on 100B tokens, while matching or exceeding DoReMi using just 10% of the computational resources. Our experiments also show that (1) Data mixtures significantly impact performance; (2) Web corpora rather than data perceived as high-quality like Wikipedia have the strongest positive correlation with downstream performance; (3) Domains interact in complex ways often contradicting common sense, thus automatic approaches like REGMIX are needed; (4) Data mixture effects transcend scaling laws. Our code is available at https://github.com/sail-sg/regmix.

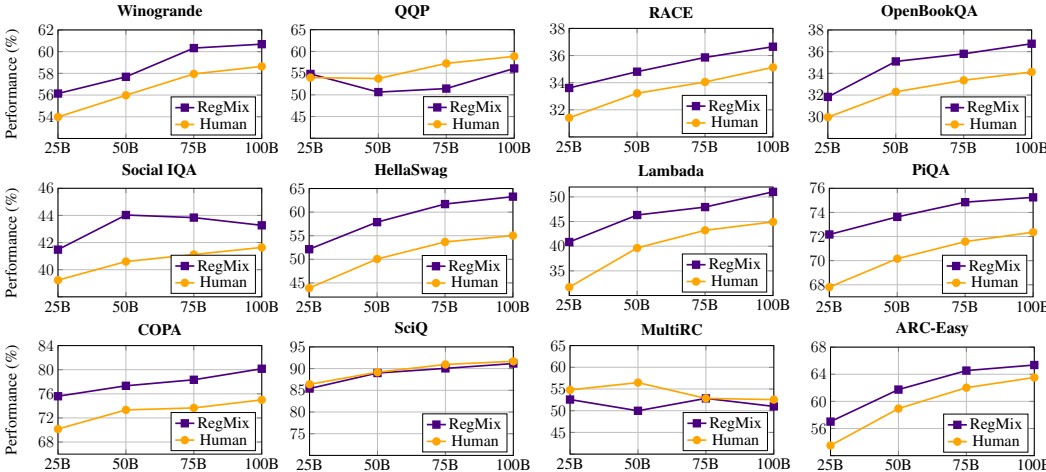

Figure 1: Performance comparison on 7B parameter models between Human selection and REGMIX across various tasks and training token quantities (e.g., up to 100B). REGMIX demonstrates consistent performance improvements over Human across varying data scales for the majority of tasks evaluated. In particular, for tasks exhibiting positive scaling behavior, the performance benefit is preserved with increased training data, suggesting robust scaling properties of REGMIX.

---

*The first two authors contributed equally.

# 1 INTRODUCTION

The availability of large-scale public datasets has been a key factor enabling the creation of large language models (LLMs). Most data is available on the Internet and includes academic papers (e.g. arXiv), books (e.g. Project Gutenberg), and code (e.g. GitHub). For the creation of one of the first LLMs, GPT-3 (Brown et al., 2020), the authors had already recognized the importance of selecting the best data for training, and thus they decided to upsample Wikipedia due to its perceived high quality. However, such manual data selection is not scalable and may lead to a suboptimal selection (Albalak et al., 2024). As the size and diversity of data used for LLM pre-training continue to grow, determining the optimal data mixture becomes increasingly challenging. It gives rise to the critical research question: *How can we select the optimal data mixture in a scalable and efficient manner?*

Prior work (Xie et al., 2023a; Fan et al., 2023; Albalak et al., 2023) employs small-scale models ("proxy models") to predict the domain weights for large-scale language models. These works train proxy models with a substantial number of tokens (e.g., 100B), sometimes even the same number as used for training LLMs, and dynamically adjust the data allocation strategy by monitoring the training dynamics. However, these approaches become inefficient as the training data used for pre-training LLMs continues to grow. Training a proxy model for current models, such as Llama-3, would require using up to 15T tokens (AI, 2024), which is likely too expensive and too slow to make it worthwhile.[1]

In this work, we argue that *training small models on a limited set of tokens* is sufficient to predict an effective data mixture for LLM training. Our key assumption is the *rank invariance of data mixtures*, which posits that the relative ranking of data mixtures in terms of their impact on model performance is consistent across different model sizes and numbers of training tokens. Under this assumption, the key challenge lies in discovering the top-ranked data mixture from the near-infinite number of potential data mixtures. To do so, we treat the data mixture selection as a regression task. Rather than exhaustively training small models with every possible mixture, we train only a set of small models, each with a unique data mixture. Based on the performance of these models and their mixtures, we fit a regression model to predict the performance of other data mixtures. Our approach is significantly more scalable than prior work, as it allows for parallel training of small proxy models rather than training a single model for a long time. Further, the regression model provides insights into domain interactions that can facilitate understanding and data curation.

To validate REGMIX, we train models with 1M and 1B parameters[2] with different data mixtures. By training 512 models with 1M parameters on 1B tokens,[3] we are able to predict the optimal data mixture among 64 models that are $1000\times$ larger (1B parameters) and trained $25\times$ longer (25B tokens) as depicted in Figure 2. Moreover, the optimized data mixture using REGMIX yields a better model than human selection, and achieves performance on par with the flagship DoReMi method (Xie et al., 2023a) despite it requiring less total compute and allowing for parallel training. We also find that (1) Data mixture significantly impacts downstream performance, resulting in substantial differences of up to 14.6% in single-task performance; (2) General web corpora (e.g., CommonCrawl), rather than Wikipedia, exhibit the strongest positive correlation with improved performance across downstream tasks; (3) The interactions between domains are complex and often contradict intuition, highlighting the need for automated approaches like REGMIX; (4) Data mixture effects transcend scaling laws, and REGMIX captures the complexity by considering all domains together.

# 2 RELATED WORK

**Data selection and mixture** is concerned with curating data to optimize some goals, usually model performance (Koh & Liang, 2017; Albalak et al., 2024). Prior methods can be categorized into: **(1) Token-level** selection is the most fine-grained level of selection dealing with the filtering of tokens (Lin et al., 2024). **(2) Sample-level** selection is about choosing individual training examples. It is commonly employed for selecting fine-tuning data (Thakkar et al., 2023; Das & Khetan, 2023; Xie et al., 2023b; Engstrom et al., 2024; Xia et al., 2024; Liu et al., 2024; Bukharin & Zhao, 2023; Kang et al., 2024; Mekala et al., 2024; Sachin Parkar et al., 2024; Yang et al., 2024b). For the pre-training of LLMs, most methods rely on heuristics (Rae et al., 2021; Sharma et al., 2024; Soldaini et al., 2024;

---

[1]These approaches often suffer from instability issues. Details can be found in Appendix F.

[2]Our model sizes mentioned in this paper refer to the number of non-embedding parameters, as embedding parameters account for a disproportionately large portion in smaller models.

[3]The estimated FLOPs for training $512\times$ 1M models is nearly 2% of the FLOPs required for one 1B model.

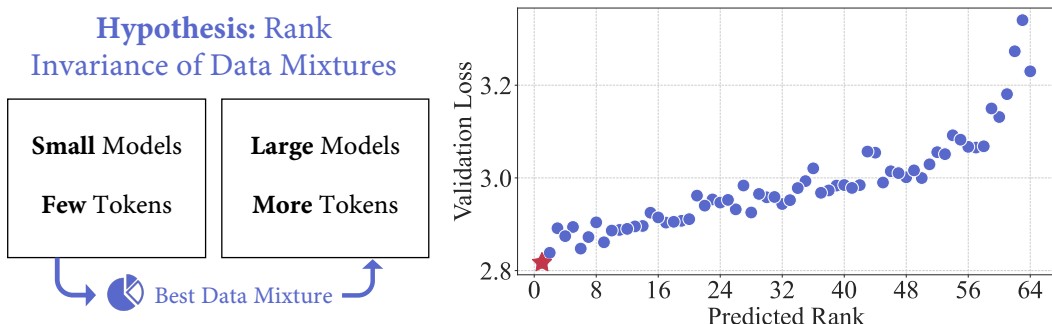

Figure 2: **Left**: We hypothesize the rank invariance of data mixtures across model sizes and numbers of training tokens. Leveraging this hypothesis, we use small models trained on fewer tokens to predict the effective data mixture for training large models with substantially more tokens. **Right**: By training $512\times$ 1M models, our method identifies the best data mixture prior to training $64\times$ 1B models. The predicted best data mixture, denoted by the red star, achieves the lowest validation loss.

Li et al., 2024), but there have been some learned approaches using optimization algorithms (Chen et al., 2024; Mindermann et al., 2022; Shao et al., 2024; Yu et al., 2024), model perplexity (Marion et al., 2023; Muennighoff et al., 2023; Ankner et al., 2024), or LLMs to inform the sample selection process (Wettig et al., 2024; Sachdeva et al., 2024; Zhang et al., 2024b). **(3) Group-level** selection assumes the data can be grouped into pools that are then optimally mixed. While early work again relies on manual mixtures (Gao et al., 2021; Brown et al., 2020), learned mixtures have become more common (Albalak et al., 2024). Learned approaches either leverage proxy models to determine fixed weights for each group ("offline selection") (Rae et al., 2021; Xie et al., 2023a; Fan et al., 2023) or dynamically adjust the weights during training of the final model ("online selection") (Wang et al., 2020; Chen et al., 2023; Albalak et al., 2023). Our approach, REGMIX, is an offline group-level selection method. Different from the flagship algorithm in this category, DoReMi (Xie et al., 2023a), REGMIX does not require training a single model for hundreds of thousands of steps, but instead a few small models for short durations. As these can be trained in parallel, our approach is more scalable, while also yielding better weights leading to a more performant final model.

**Data scaling laws** explore interactions of data quantity, quality, and mixing proportions, as LLMs are scaled up. Muennighoff et al. (2023) introduce scaling laws for data-constrained scenarios and Goyal et al. (2024) try to extend this approach to deal with multiple data pools. Prior research has confirmed that different datasets require different scaling (Hoffmann et al., 2022; Pandey, 2024), thus Ye et al. (2024) and Ge et al. (2024) propose functional relationships to predict the impact of mixtures on language modeling loss. Some work has investigated optimal mixtures during continued pre-training rather than from scratch training (Que et al., 2024; Dou et al., 2024). While most of these works focus on validation loss, others investigate downstream performance and develop predictive relations with loss (Gadre et al., 2024; Yang et al., 2024a; Xia et al., 2022). Different from data scaling works that attempt to find an analytical scaling function (Hoffmann et al., 2022), REGMIX directly optimizes the target metric using regression models. REGMIX is designed for from-scratch pre-training. In line with previous research (Huang et al., 2024), we also find strong correlations between loss and downstream performance, especially for loss on web corpora.

## 3 REGMIX: DATA MIXTURE AS REGRESSION

As illustrated in Figure 3, our method involves four key steps: (1) Generate random data mixtures and train small-scale proxy models on these mixtures. (2) Fit a linear regression model using the mixtures as features and the target value as the label. (3) Simulate the data mixture space on a larger scale and leverage the regression model to identify the best mixture for the target value. (4) Train a large-scale model using the simulated best data mixture.

### 3.1 TRAIN SMALL-SCALE PROXY MODELS

The first step is to train a set of small-scale proxy models on multiple different data mixtures. To reduce the required runs, we aim to select a diverse range of data mixtures that cover extreme weights from 0% to 100% for each domain. We achieve this by using a Dirichlet distribution based on the token distribution, which allows us to sample a wide range of values and expose the regression

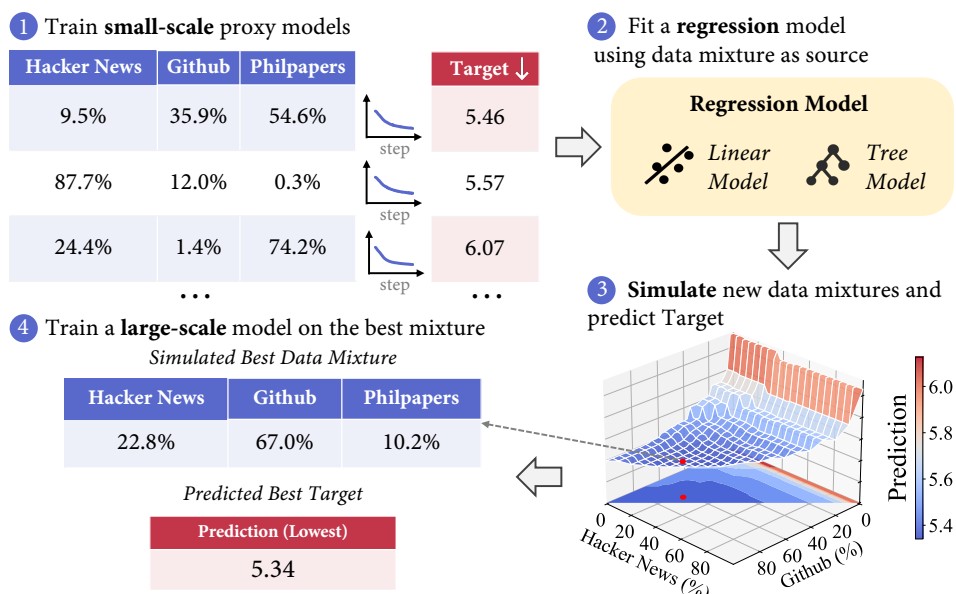

Figure 3: The illustration of our method using Hacker News, GitHub, and Philpapers as training domains, with the loss on the StackExchange domain as the Target (where ↓ indicates lower is better). A regression model is fitted using small-scale proxy model training logs and employed to predict the best data mixture within the simulation space, enabling direct prediction of the data mixture for large-scale language model pre-training. Note that the Philpapers domain is omitted in the simulation plot (3) for simplicity.

Table 1: Overview of the Pile dataset (Gao et al., 2021) with datasets that are no longer available due to copyright issues marked in gray. In our experiments, we use the 17 available domains to study the data mixture for language model pre-training.

| Component | Effective Size | Component | Effective Size |
|---|---|---|---|
| Pile-CC | 227.12 GiB | OpenSubtitles | 19.47 GiB |
| PubMed Central | 180.55 GiB | Wikipedia (en) | 19.13 GiB |
| Books3 | 151.44 GiB | DM Mathematics | 15.49 GiB |
| OpenWebText2 | 125.54 GiB | Ubuntu IRC | 11.03 GiB |
| ArXiv | 112.42 GiB | BookCorpus2 | 9.45 GiB |
| Github | 95.16 GiB | EuroParl | 9.17 GiB |
| FreeLaw | 76.73 GiB | HackerNews | 7.80 GiB |
| Stack Exchange | 64.39 GiB | YoutubeSubtitles | 7.47 GiB |
| USPTO Backgrounds | 45.81 GiB | PhilPapers | 4.76 GiB |
| PubMed Abstracts | 38.53 GiB | NIH ExPorter | 3.79 GiB |
| Gutenberg (PG-19) | 27.19 GiB | Enron Emails | 1.76 GiB |

models to various extremes. Simultaneously, basing the distribution on the token distribution ensures that the overall data mixture statistically reflects the availability of data. For example, this prevents any single domain with a token count below 1% from being overly emphasized, which is not feasible for large-scale training since there are not enough available tokens from that domain. In practice, we multiply the token distribution by a value from 0.1 to 5.0 to construct various sparse and near-uniform distributions, then use these distribution vectors as the Dirichlet distribution hyperparameter $\alpha$.

After training small-scale proxy models for a few steps, we can obtain several well-trained small models. For example, in our main experiment, each proxy model contains 1M parameters and is trained on 1B tokens. We can then choose to evaluate these trained models on domains or benchmarks to get the target value we want to optimize. Generally, the target value can be the loss on a domain, as shown in Figure 3 for the StackExchange domain. Once we have obtained these target values, we can use the data mixture as features and the target values as labels to fit a regression model.

## 3.2 FIT A REGRESSION MODEL

The second step is to fit a regression model using the data mixture as features, and the target value as labels. The regression task is a conventional supervised learning task that involves predicting a continuous target variable $y$ based on input features $X = (x_1, x_2, \ldots, x_n)$. The goal is to find a

function $f$ that best maps the input features to the target variable, such that $y = f(X) + \epsilon$, where $\epsilon$ represents the error or noise in the data. In the context of this paper, the input features $X$ correspond to the domain weights of the data mixture, and the target variable $y$ is the value we want to optimize. Using this data, we train regression models that learn a function to predict the target value based on arbitrary data mixtures without requiring further training.

**Linear regression.** The linear regression model is widely used in regression. It assumes a linear relationship between the input features and the target variable, which can be represented as:

$$y = \omega_0 + \omega_1 x_1 + \ldots + \omega_n x_n + \epsilon \tag{1}$$

where $\omega_0$ is the intercept, and $\boldsymbol{\omega} = (\omega_1, \ldots, \omega_n)$ are the coefficients associated with the respective input features $x_1, \ldots, x_n$. The coefficients $\boldsymbol{\omega}$ are typically estimated using techniques such as ordinary least squares, aiming to minimize the sum of squared residuals between the predicted and actual target values. In practice, we employ linear regression with L2 regularization, also known as ridge regression, which applies a penalty to the magnitude of $\boldsymbol{\omega}$ to prevent overfitting.

**LightGBM regression.** LightGBM (Ke et al., 2017) is a powerful gradient-boosting algorithm that can be used for both regression and classification tasks. In the context of regression, LightGBM learns an ensemble of decision trees to predict the target variable. The process is guided by a gradient-based optimization algorithm, which minimizes a specified loss function (e.g. mean squared error). Moreover, LightGBM is designed to be efficient and scalable, making it suitable for large datasets.

### 3.3 SIMULATE AND PREDICT

Once we have trained the regression model, we can efficiently explore the entire space of possible data mixtures. By using the trained model to predict the target value for each potential data mixture, we can quickly identify the input that yields the best target value. This simulation-based optimization is relatively cheap, as both the simulation and the regression prediction are computationally fast. For example, running prediction for 1,000,000 data mixtures takes less than 10 CPU seconds.

### 3.4 LARGE-SCALE MODEL TRAINING

After identifying the best data mixture with simulation, we generalize the top-ranked data mixture to a large-scale model training with many more tokens. As shown in Figure 3, we directly use the best data mixture for training the larger model. In practice, to increase the robustness of our regression prediction, we select the top 100 mixtures and average them as the data mixture for large-scale training.

## 4 EVALUATING ON REGRESSION PREDICTION

In this section, we evaluate the ability of REGMIX to predict the effect of unseen data mixtures. First, we fit the regression model using training artifacts of small (i.e., 1M parameter) models and evaluate the loss prediction performance on small models. Then, to verify our *rank invariance hypothesis*, we test the learned regression on predicting the rank across model sizes and the number of tokens.

### 4.1 EXPERIMENTAL SETUP

**Datasets and models.** We conduct our experiments using the domains of the Pile dataset (Gao et al., 2021) depicted in Table 1. Due to copyright concerns, we utilize the 17 subsets available on HuggingFace [4] that do not violate copyright issues. We consider both linear and LightGBM regression models, where the target $y$ is set to be the validation loss of the Pile-CC domain.

**Training and evaluation.** The regression model is fitted using the training artifacts of $512\times$ 1M models with 1B tokens, and evaluated on $256\times$ unseen data mixtures for 1M, 60M models (each trained with 1B tokens) and $64\times$ unseen data mixtures for 1B models (each trained with 25B tokens).

**Evaluation metrics.** We use two metrics to benchmark our regression models: (1) *Spearman Rank Correlation ($\rho$)* is a non-parametric measure of the strength and direction of the association between two ranked variables. (2) *Mean Squared Error (MSE)* is a common metric used to evaluate the model by measuring the average squared differences between the predicted and actual values.

---

[4]https://huggingface.co/datasets/monology/pile-uncopyrighted

Table 2: We fit the regression model based on the results of the $512\times$ 1M parameter models trained on 1B tokens, and evaluate it on **unseen data mixtures** for 1M, 60M, and 1B parameter models depicted below. The Spearman correlation $\rho$ compares the predicted and actual ranks, while MSE measures the loss prediction performance. *1M* refers to models with 1M parameters trained on 1B tokens, *60M* refers to models with 60M parameters trained on 1B tokens, and *1B* refers to models with 1B parameters trained on 25B tokens. Due to scale differences, MSE values for 60M and 1B models are not directly comparable and are omitted from analysis.

| Test On | *1M* | | *60M* | *1B* |
|---------|------|------|-------|------|
| **Method** | $\rho$ ($\uparrow$) | **MSE** ($\downarrow$) | $\rho$ ($\uparrow$) | $\rho$ ($\uparrow$) |
| Linear | 90.08 | 0.13 | 89.26 | 88.01 |
| LightGBM | 98.45 | 0.04 | 98.64 | 97.12 |

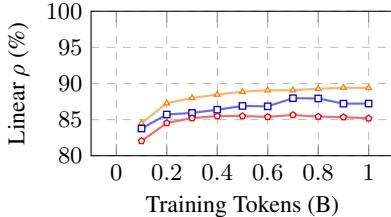 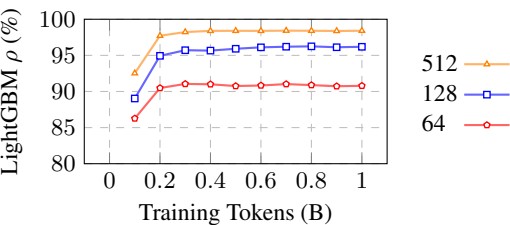

Figure 4: The plot of Spearman Rank Correlation $\rho$ between the predicted ranks and true ranks of Linear regression (**Left**) and LightGBM regression (**Right**) across different training tokens and different number of proxy models. As shown, increasing the number of proxy models significantly boosts $\rho$, while adding more training tokens has diminishing returns.

## 4.2 EXPERIMENTAL RESULTS

**High correlation across model sizes.** As shown in Table 2, the LightGBM model demonstrates superior performance over linear regression models across all three metrics, with its advantage becoming increasingly pronounced when evaluating on larger models with more training tokens. Meanwhile, the fact that 1M models trained with 1B tokens can achieve such a high correlation of 97.12% on unseen mixtures of 1B models with 25B tokens directly validates our *rank invariance hypothesis*.

**Proxy model count outweighs training token count.** Given the same FLOPs budget for small-scale training, we can either increase the token count (i.e., the number of training tokens) or the number of proxy models. Therefore, we study which approach would yield better performance. As shown in Figure 4, increasing the training tokens of the proxy models saturates after approximately 0.25B tokens. In contrast, increasing the number of proxy models consistently enhances performance, particularly for the LightGBM model. Notably, the performance of 512 models trained on 0.2B tokens surpasses that of 128 models trained on 0.8B tokens, indicating that increasing the number of proxy models is more effective than increasing the training token count beyond a certain token threshold.

## 5 EVALUATING ON DOWNSTREAM TASKS

In this section, we apply our method to demonstrate its effectiveness on realistic downstream tasks. For evaluation, we exclude specific benchmarks that exhibit large performance variance (e.g., RTE) according to the performance traces reported in previous work (Mehta et al., 2024) and our observations during pre-training. Ultimately, we select the following benchmarks as our downstream tasks: Social IQA (Sap et al., 2019), HellaSwag (Zellers et al., 2019), PiQA (Bisk et al., 2020), OpenBookQA (Mihaylov et al., 2018), Lambada (Paperno et al., 2016), SciQ (Welbl et al., 2017), ARC Easy (Clark et al., 2018), COPA (Sarlin et al., 2020), RACE (Lai et al., 2017), LogiQA (Liu et al., 2020), QQP (Wang et al., 2018), WinoGrande (Sakaguchi et al., 2021), and MultiRC (Khashabi et al., 2018). These benchmarks cover a diverse range of tasks, enabling a comprehensive evaluation of the real-world impact of REGMIX. For each benchmark, we use normalized accuracy as the evaluation metric if provided by lm-eval-harness (Gao et al., 2023) else we use regular accuracy.

Table 3: We experiment with 64 models, each with 1B parameters trained on different data mixtures, and evaluate their performance across various benchmarks. The reported performance on each task is the average score from 0-shot to 5-shot settings, following Muennighoff et al. (2023). Here, we present the worst and best model performances on each task, and detailed experimental results for individual models can be found in Appendix G.

| Benchmark | Worst Model | Best Model | Δ |
|---|---|---|---|
| Social IQA (Sap et al., 2019) | 32.4 | 33.9 | 1.5 |
| HellaSwag (Zellers et al., 2019) | 33.0 | 43.4 | 10.4 |
| PiQA (Bisk et al., 2020) | 60.2 | 69.0 | 8.8 |
| OpenBookQA (Mihaylov et al., 2018) | 25.8 | 31.2 | 5.4 |
| Lambada (Paperno et al., 2016) | 18.9 | 33.5 | 14.6 |
| SciQ (Welbl et al., 2017) | 76.7 | 82.9 | 6.2 |
| ARC Easy (Clark et al., 2018) | 44.9 | 52.2 | 7.3 |
| COPA (Sarlin et al., 2020) | 61.5 | 70.5 | 9.0 |
| RACE (Lai et al., 2017) | 27.9 | 32.5 | 4.6 |
| LogiQA (Liu et al., 2020) | 23.2 | 27.7 | 4.5 |
| QQP (Wang et al., 2018) | 48.0 | 59.7 | 11.7 |
| WinoGrande (Sakaguchi et al., 2021) | 50.3 | 53.2 | 2.9 |
| MultiRC (Khashabi et al., 2018) | 47.6 | 55.7 | 8.1 |
| Average Performance | 43.7 | 47.9 | 4.2 |

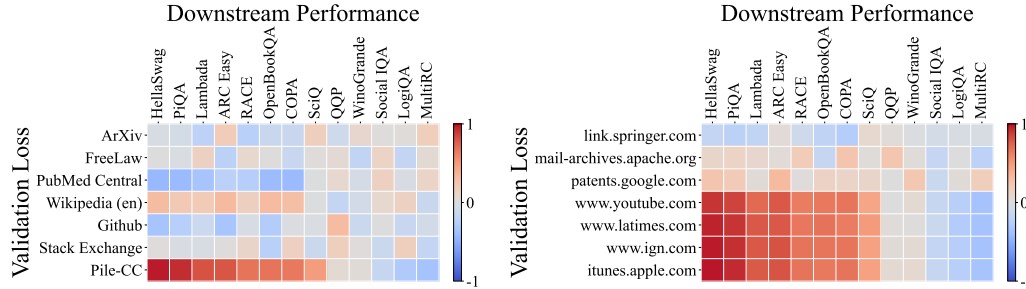

(a) Correlation between validation loss by domains of the Pile and downstream performance.

(b) Correlation between validation loss by URL domain within the Pile-CC subset and downstream performance.

Figure 5: The correlation between validation losses across domains and downstream performance for the $64 \times$ 1B models. Note that we take the negative of the loss value when calculating the correlation, as this makes the visualization more intuitive. The same applies for Figure 7.

## 5.1 DATA MIXTURE SIGNIFICANTLY IMPACTS DOWNSTREAM PERFORMANCE

Initially, we train 64 models, each with 1B parameters, using different data mixtures. Every model is trained on 25B tokens[5] from the Pile dataset (Gao et al., 2021), with tokens allocated based on their corresponding domain weights. Table 3 presents the performance of the worst and best models on each downstream task. The reported performance is the average from 0-shot to 5-shot evaluations, scored using the lm-eval-harness evaluation framework (Gao et al., 2023; Biderman et al., 2024). We find that the data mixture significantly impacts downstream performances, with the largest performance $\Delta$ reaching 14.6 on the Lambada task. This underscores the importance of studying the optimal data mixture.

## 5.2 WEB CORPORA BENEFITS DOWNSTREAM PERFORMANCE THE MOST

Next, we visualize the correlation between the validation losses of our 64 1B models across different domains and their performance on various downstream tasks in Figure 5 (a). Prior to visualization, we hypothesized that the validation loss on the Wikipedia (en) subset would exhibit a strong correlation with most downstream tasks, as it is a high-quality dataset, and many downstream tasks are derived

---

[5]We set the token quantity such that it is compute-optimal according to Chinchilla (Hoffmann et al., 2022).

Table 4: Performance comparison of different data selection methods. Human refers to the weights put forth in The Pile (Gao et al., 2021), Pile-CC to only training on the Pile-CC component, PPL to using the perplexity filtering methods from Ankner et al. (2024), ODM to the weights from Albalak et al. (2023) and DoReMi to the weights from Xie et al. (2023a). The reported performance on each task is the average score from 0-shot to 5-shot settings, while the highest score on each task is highlighted in bold. We estimate the compute (measured in FLOPs) required to arrive at the training data mixture.

| Benchmark | Human | DoReMi | PPL | ODM | Pile-CC | REGMIX |
|---|---|---|---|---|---|---|
| Social IQA (Sap et al., 2019) | 33.6 | 33.4 | 33.3 | 33.7 | 33.2 | **33.8** |
| HellaSwag (Zellers et al., 2019) | 37.4 | 43.4 | 43.1 | 37.2 | 44.1 | **44.2** |
| PiQA (Bisk et al., 2020) | 65.0 | 68.3 | 68.5 | 64.4 | 69.2 | **69.3** |
| OpenBookQA (Mihaylov et al., 2018) | 28.2 | 30.3 | 30.3 | 30.0 | **31.1** | 30.3 |
| Lambada (Paperno et al., 2016) | 29.8 | 32.1 | **35.4** | 29.6 | 33.2 | 34.2 |
| SciQ (Welbl et al., 2017) | 80.1 | 81.6 | 78.6 | 79.8 | 81.8 | **82.8** |
| ARC Easy (Clark et al., 2018) | 49.4 | 50.6 | 50.5 | 47.9 | **51.8** | 51.7 |
| ARC Challenge (Clark et al., 2018) | 26.3 | 26.1 | 25.9 | 25.6 | **26.7** | 25.7 |
| COPA (Sarlin et al., 2020) | 66.7 | 68.5 | 69.2 | 68.2 | 65.8 | **70.2** |
| RACE (Lai et al., 2017) | 29.0 | 31.3 | 31.5 | 29.7 | **31.8** | 31.3 |
| LogiQA (Liu et al., 2020) | 25.5 | 26.4 | 27.5 | 25.6 | **27.6** | 25.8 |
| QQP (Wang et al., 2018) | 52.4 | 56.6 | 50.0 | 53.1 | 57.0 | **58.3** |
| WinoGrande (Sakaguchi et al., 2021) | **53.1** | 52.2 | 52.8 | 51.8 | 52.1 | **53.1** |
| MultiRC (Khashabi et al., 2018) | **54.3** | 53.8 | 50.4 | 53.3 | 50.3 | 51.7 |
| Estimated FLOPs | 0 | 3.7e19 | 1.8e19 | 0 | 0 | 3.5e18 |
| Average Performance | 45.1 | 46.8 | 46.2 | 45.0 | 46.8 | **47.3** |
| Best On | 2 / 14 | 0 / 14 | 1 / 14 | 0 / 14 | 5 / 14 | **7 / 14** |

from Wikipedia text. Similarly, previous work often takes WikiText (Merity et al., 2016) as a standard benchmark to indicate language model performance.

However, surprisingly, the validation loss on the Pile-CC dataset shows the strongest correlation with most downstream tasks. For instance, the correlation coefficient between the HellaSwag task and the Pile-CC validation loss is remarkably close to $1.0$. This unexpected result challenges the conventional assumption that WikiText is the most representative dataset for evaluating LLMs. Furthermore, this result aligns with the findings of previous studies (Gadre et al., 2024; Huang et al., 2024), which discovered that the validation loss on the web dataset closely relates to downstream performance.

Moreover, we analyze the correlation between the loss of models on the C4100Domain validation set (Magnusson et al., 2023), which is taken from the C4 dataset (Raffel et al., 2019) and supposed to share a similar distribution as Pile-CC since they are all derived from the CommonCrawl corpus. Since CommonCrawl is a collection of diverse domains, we would expect the correlation between the loss of each domain and the downstream tasks to vary. However, surprisingly more than 85% of the domains exhibit a very strong correlation with Pile-CC (full correlation graph in Appendix D). This is exemplified by the www.ign.com domain, which closely mirrors the overall correlation graph of Pile-CC, as illustrated in Figure 5 (b). It also suggests that the high correlation between Pile-CC and downstream task performance may be attributed to its *diverse coverage across various topics and domains*.

### 5.3 DATA MIXTURE BY REGMIX IMPROVES DOWNSTREAM PERFORMANCE

Previous work has shown that the data mixture method can accelerate LLM pre-training by achieving a smaller validation loss (or perplexity) using less training tokens (Xie et al., 2023a). However, a critical challenge is determining *which validation loss to optimize*. While the intuitive approach is to minimize loss across all domains, our analysis of 1M training logs reveals significant practical challenges. Therefore, instead of pursuing a broad optimization strategy, we strategically focus on minimizing validation loss on Pile-CC, which allows for meaningful progress by leveraging Pile-CC which most consistently predicts overall model performance on downstream tasks.

We implement two approaches to determine the data mixture. The first approach relies on human intuition. Since Pile-CC and its own distribution should be the closest match, we hypothesized that pre-training solely on Pile-CC might yield better performance than baselines. The second approach leverages REGMIX, using the Pile-CC validation loss as the target variable. We employed LightGBM to predict the data mixture which can minimize the Pile-CC validation loss.

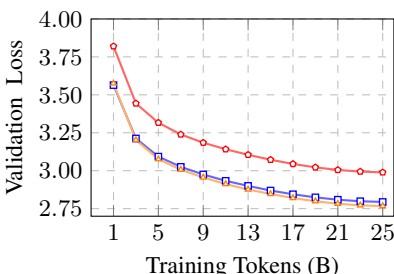 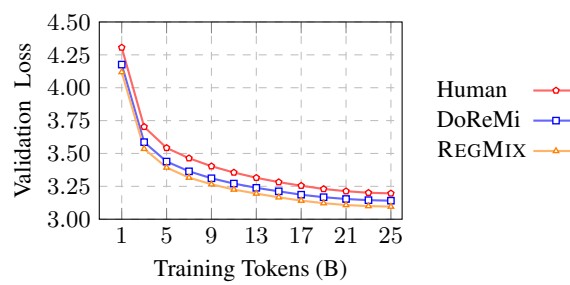

Figure 6: **Left**: The validation loss on Pile-CC of different methods with Pile-CC in the pre-training corpus. **Right**: The validation loss on Pile-CC excluding Pile-CC in the pre-training.

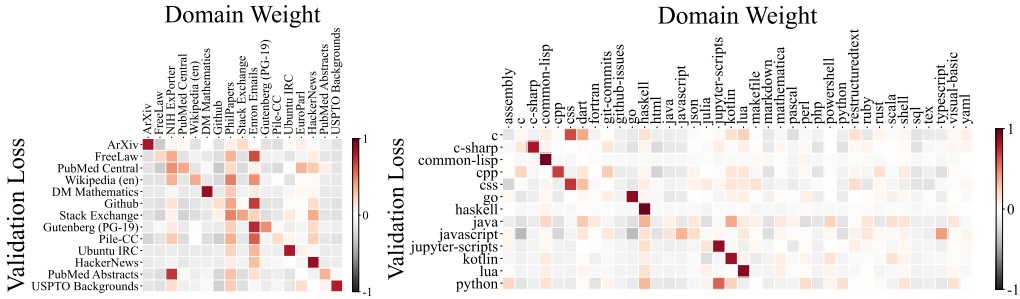

Figure 7: The visualization of correlations between different target domain validation losses and training domain weights using the linear regression model. **Left** is on the Pile dataset, and **Right** is on the Stack dataset. A high correlation indicates that increasing the training domain weight has a positive impact on reducing the target domain validation loss.

We evaluate our proposed approaches against robust benchmarks, including human-curated selections for the Pile corpus (Gao et al., 2021), a sample-level perplexity-based filtering method (PPL) (Ankner et al., 2024), an online group-level method Online Data Mixing (ODM) (Albalak et al., 2023), and the flagship group-level method DoReMi (Xie et al., 2023a). For ODM and DoReMi, we obtain the data mixture directly from their reported best domain weights and re-normalize it across the available 17 domains. This may result in sub-optimal performance for them compared to the originally reported results. As shown in Table 4, both Pile-CC Only and REGMIX demonstrate strong performance compared to the baselines. On the widely used HellaSwag benchmark, REGMIX shows an improvement of 6.8 over Human selection. Additionally, REGMIX beats all other three methods on the task performance in 7 out of 14 cases and yields the highest average score. The surprisingly strong performance of Pile-CC Only reinforces the conclusion from our previous section: web corpora benefits on downstream performance. Finally, REGMIX surpasses the Best Model in Table 3, demonstrating that our automatic data mixture approach is more efficient than random search.

While the Pile-CC validation loss is an informative indicator for downstream performance, it may not generalize to every task of interest. Sometimes we may not be able to assume that the validation set stems from a similar data distribution as the training set, but rather face an out-of-distribution scenario. To verify the effectiveness of our method in out-of-distribution scenarios, we fully exclude the Pile-CC domain from the pre-training corpus and use the remaining domains to find the optimal data mixture that minimizes Pile-CC validation loss. As illustrated in Figure 6 (right), our proposed method still outperforms baseline approaches. This demonstrates that REGMIX is robust regardless of whether the target domain is in- or out-of-distribution. We additionally provide the results of regression evaluation under this setting in Figure 6.

### 5.4 DOMAIN INTERACTIONS ARE CHALLENGING FOR HUMANS TO UNDERSTAND

To understand the impact of different domains on each other, we visualize the coefficients ($\omega$) of the linear regression model in Figure 7. The visualization provides insights into how the various data domains contribute to the others, revealing complex interactions among them. We also display code correlation diagrams for each 1M code model trained on The Stack dataset (Kocetkov et al.,

Table 5: Performance Comparison on 100 Domains for 1M and 60M Models. This table compares the rank correlation ($\rho$, higher is better) and mean squared error (MSE, lower is better) for linear and LightGBM regression models. We train 1,000 runs for 1M models, fit the regression model, and verify the rank correlation between the ranking predicted by the regression model and the ground-truth ranking of 64 unseen data mixtures on both 1M and 60M models.

| Test On | *1M* | | *60M* |
|---|---|---|---|
| **Method** | $\rho$ (↑) | **MSE** (↓) | $\rho$ (↑) |
| Linear | 90.33 | 0.12 | 88.64 |
| LightGBM | 99.53 | 0.02 | 98.80 |

2022). Surprisingly, both the domain interaction visualization and the code correlation diagrams display complex relationships that are difficult for human experts to fully comprehend. For example, the PhilPapers domain in the Pile dataset appears to provide gains for all other domains under the linear regression modeling, which is a non-obvious finding that challenges intuitive human understanding. These visualizations highlight the inherent complexity in determining the optimal data mixture, underscoring the value of our automated REGMIX approach in efficiently identifying high-performing mixtures, rather than relying solely on human intuition.

## 5.5 DATA MIXTURE EFFECTS TRANSCEND SCALING LAWS

Recent research (Ye et al., 2024; Ge et al., 2024) has demonstrated the feasibility of scaling laws for data mixture. However, our findings in Section 5.4 suggest that the relationship between domain weights and validation loss is more complex than scaling laws might imply. To visualize this complexity, we plotted all experimental points of our 1M training logs in Figure 9. If the scaling law of data mixture held true, we would expect to see a clear log-log linear relationship across all domains. However, our results reveal a more nuanced picture. For example, the DM Mathematics domain, possibly due to its distinct distribution compared to other domains, exhibits a near log-log linear relationship between loss and domain weight. In contrast, for most domains like Pile-CC show more complex patterns, where predicting validation loss is non-trivial. As shown, domain interactions appear to be intricate, making it challenging to predict the validation loss for a domain based solely on its weight in the mixture. These findings suggest that while scaling laws provide valuable insights, they may not fully capture the intricacies of data mixture dynamics. Our approach addresses the challenge by modeling the entire data mixture as input for the regression model, providing a more comprehensive framework for predicting the validation loss while simultaneously accounting for all domain weights.

## 5.6 EXTEND REGMIX TO 100 DOMAINS

To demonstrate REGMIX's scalability, we conducted preliminary experiments with 100 finer-grained domains. One key challenge lies in clustering web content into meaningful domain representations. Intuitively, we define domains by base URLs from the FineWeb dataset (Penedo et al., 2024a), chosen based on token availability. Example domains include *articles.latimes.com*, *blogs.wsj.com*, *en.wikipedia.org*, *everything2.com*, and *techcrunch.com*.

We train 1,000 small-scale models (1M parameters) across different data mixtures, use the training runs to fit a regression model and then predict the data mixture for models with 1M and 60M parameters. Rank correlation ($\rho$) and mean squared error (MSE) were evaluated for both linear and LightGBM regression models as shown in Table 5. The results demonstrate the effectiveness of REGMIX when extending to 100 domains.

## 6 CONCLUSION

In this paper, we present REGMIX, a novel approach for automatically selecting high-performing data mixtures for pre-training large language models. By formulating the data mixture problem as a regression task, REGMIX trains small models to predict the impact of different mixtures, enabling efficient identification of the optimal combination. We demonstrate REGMIX's effectiveness by predicting the best data mixture among 64 1B-parameter models. Our large-scale study provides insights into data mixture impacts, the relationship between loss and downstream performance, and the challenges faced by human experts in determining optimal data mixtures.

ETHICS STATEMENT

Optimizing the data mixture for LLM pre-training raises several ethical issues. First, the optimized data mixture might be biased toward certain domains, which is good for achieving better performance. However, certain domains might be underrepresented or misrepresented, leading the trained models to perform poorly or produce biased results for these domains. Second, though our method aims to optimize the data mixture efficiently, searching for the optimal data mixture still requires computational resources, leading to high energy consumption and environmental impact. It is worthwhile to explore how to further reduce the computation cost.

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

## A  Broader Impact

In this paper, we propose REGMIX, a novel method for optimizing data mixture for language model pre-training. A unique contribution of REGMIX is its novel use of **ultra-small** proxy models (i.e., 1M parameters) to optimize data mixtures for language model pre-training, an approach previously unexplored in the field. This novel use of ultra-small proxy models reduces computational overhead to less than 2% of final training costs, dramatically lowering barriers for data mixture research within academic budgets. Through the focus on computational efficiency and our commitment to open science (i.e., with all datasets and trained models publicly available), we believe REGMIX represents a significant advancement toward democratizing research in language model pre-training.

Beyond the methodology contribution, our work delivers several novel empirical insights in data mixture research. We provide the first comprehensive demonstration of significant performance variations across different data mixtures, supported by extensive experiments with 1B-parameter models trained on 64 distinct mixtures and rigorously evaluation across 12 benchmarks. Our results establish the superiority of automatic data mixture optimization over human intuition-based approaches, with PhilPapers serving as an interesting in-depth case study illustrating how domain interactions follow complex patterns that transcend human intuition.

## B  Limitations

Despite making progress in understanding and optimizing data mixtures for better performance, our method still has several limitations.

**The rank invaraince assumption.**   Our investigation of the rank invariance assumption currently focuses on model scales from 1M to 1B parameters. While we aimed to verify the hypothesis at larger scales, establishing statistically meaningful correlations for 3B models would require training 64 different models with 50B tokens each, equivalent to training one 3B model on 3.2T tokens, which significantly exceeds our computational resources.

**The maximum model parameters.**   We have verified that small models can be used to predict the optimal data mixture for large-scale runs with up to 1B parameters. However, much larger models are commonly trained with 7B or 70B parameters (Touvron et al., 2023). Due to compute constraints we leave the verification of REGMIX at larger scales to future work.

**The benchmark coverage.**   Owing to the scarcity of relevant data in the Pile corpus and the relatively small size of our model at 1B scale, their performance on the MMLU benchmark (Hendrycks et al., 2021) is nearly random and negligible on GSM8K (Cobbe et al., 2021). Consequently, we do not compute the correlation between the validation loss and scores on these challenging benchmarks.

**The infinite data assumption.**   Most existing data mixing methods assume the availability of unlimited data for each domain. Although we consider this issue in our no Pile-CC experiments in Section 5.3, systematically incorporating the effect of available data into the method remains challenging. Combining our method with the decay coefficient of data reuse proposed in Muennighoff et al. (2023) could be an interesting future work to explore, potentially addressing the limited data availability scenario.

**The domain assumption.**   A common assumption of existing data mixture methods (including ours) is that the domain each example belongs to is known. However, this may not always be the case and the domain needs to be obtained first. Assigning examples to domains is a hard task, which may make it challenging to apply our methods when the domain boundaries are unclear.

**The tokenizer assumption.**   All existing data mixture methods require the use of proxy models to obtain domain weights. However, a fundamental assumption of these methods is that the proxy model uses the same tokenizer and vocabulary size as the large model. Generalizing weights across different tokenizers poses significant challenges.

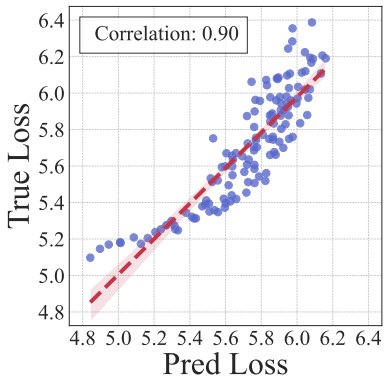 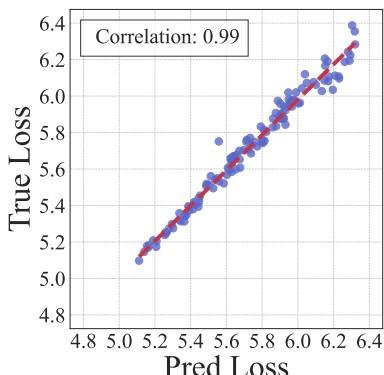

Figure 8: The visualization of loss prediction on small models (e.g., 1M parameters). **Left**: The scatter plot of predicted and true loss pairs of Linear model. **Right**: The scatter plot of predicted and true loss pairs of LightGBM model.

## C  ADDITIONAL RESULTS

### C.1  THE REGRESSION PREDICTION VISUALIZATION

As shown in Figure 8, we visualize the predicted and true loss pairs of the linear model and LightGBM model on the 1M models. The LightGBM model performs better than the linear model, achieving near 100% Spearman Rank Correlation $\rho$.

### C.2  LOSS AND RANK PREDICTION ON SMALL MODELS FOR OUT-OF-DISTRIBUTION SETTING

In Section 5, we verify the effectiveness of our method in out-of-distribution scenarios where we fully exclude the Pile-CC domain from the pre-training corpus and use the remaining domains to find the optimal data mixture that minimizes Pile-CC validation loss. We also provide the results of regression evaluation under this setting in Figure 6. Similarly, LightGBM model outperforms the linear model and achieves nearly 100% Spearman Rank Correlation $\rho$.

Table 6: The regression model is fitted using the training artifacts of $512\times$ 1M models trained with 1B tokens excluding the Pile-CC domain, and evaluated on **unseen data mixtures** for 1M parameter models. Pearson's r and MSE measure the loss prediction performance, while $\rho$ compares the predicted and actual ranks.

| Test On | 1M models with 1B tokens | |
|---|---|---|
| **Method** | $\rho$ ($\uparrow$) | MSE ($\downarrow$) |
| Linear | 83.00 | 0.08 |
| LightGBM | 95.47 | 0.04 |

### C.3  THE DERIVED DATA MIXTURES

Table 7 presents the derived data mixture weights for different methods. As illustrated, REGMIX assigns a high weight of 0.87 to the Pile-CC dataset, aligning with human intuition.

Table 7: The domain weights of different methods. In our experiments, DoReMi refers to the reported best reference model with 280M parameters and its corresponding domain weights. [†]Note that the domain weights of Human, DoReMi and Online are re-normalized from the weights reported in Xie et al. (2023a) to adapt them to the available domains. The DoReMi weight are derived from the best-performing configuration obtained using a 280M parameter model. The Online weight is derived from the final domain weights obtained by the method.

| Domain Weights | Human[†] | DoReMi[†] | Online[†] | Pile-CC | REGMIX |
|---|---|---|---|---|---|
| ArXiv | 0.134 | 0.004 | 0.0267 | 0.0 | 0.001 |
| FreeLaw | 0.049 | 0.005 | 0.0346 | 0.0 | 0.001 |
| NIH ExPorter | 0.007 | 0.008 | 0.0466 | 0.0 | 0.001 |
| PubMed Central | 0.136 | 0.006 | 0.0316 | 0.0 | 0.003 |
| Wikipedia (en) | 0.117 | 0.086 | 0.0504 | 0.0 | 0.016 |
| DM Mathematics | 0.025 | 0.002 | 0.0168 | 0.0 | 0.0 |
| Github | 0.054 | 0.022 | 0.0155 | 0.0 | 0.0 |
| PhilPapers | 0.003 | 0.034 | 0.0451 | 0.0 | 0.0 |
| Stack Exchange | 0.118 | 0.019 | 0.0353 | 0.0 | 0.0 |
| Enron Emails | 0.004 | 0.009 | 0.0228 | 0.0 | 0.002 |
| Gutenberg (PG-19) | 0.025 | 0.009 | 0.0669 | 0.0 | 0.002 |
| Pile-CC | 0.142 | 0.743 | 0.0894 | 1.0 | 0.87 |
| Ubuntu IRC | 0.009 | 0.011 | 0.0363 | 0.0 | 0.064 |
| EuroParl | 0.005 | 0.008 | 0.0315 | 0.0 | 0.0 |
| HackerNews | 0.01 | 0.016 | 0.0604 | 0.0 | 0.012 |
| PubMed Abstracts | 0.107 | 0.014 | 0.0467 | 0.0 | 0.024 |
| USPTO Backgrounds | 0.053 | 0.004 | 0.0403 | 0.0 | 0.002 |

Table 8: Performance comparison of different data selection methods using LightEval following previous work (Penedo et al., 2024b). Human refers to the weights put forth in The Pile (Gao et al., 2021), Pile-CC to only training on the Pile-CC component, and DoReMi to the weights from Xie et al. (2023a). The reported performance for each task is the average *zero-shot task performance* across five different runs, and the standard deviation. We estimate the compute (measured in FLOPs) required to arrive at the training data mixture. Scores significantly outperforming the Human baseline for each task are highlighted in **bold**, with significance determined using Cohen's d.

| Benchmark | Human | DoReMi | Pile-CC | REGMIX |
|---|---|---|---|---|
| ARC Easy (Clark et al., 2018) | $45.3 \pm 0.4$ | **46.6** $\pm 0.7$ | **47.1** $\pm 0.6$ | **47.2** $\pm 0.9$ |
| ARC Challenge (Clark et al., 2018) | $25.5 \pm 0.8$ | $25.9 \pm 0.8$ | $25.6 \pm 0.5$ | $25.6 \pm 0.5$ |
| CommonsenseQA (Talmor et al., 2019) | $31.8 \pm 1.2$ | **34.1** $\pm 0.7$ | **34.9** $\pm 0.3$ | **35.0** $\pm 0.5$ |
| HellaSwag (Zellers et al., 2019) | $36.5 \pm 0.2$ | **41.5** $\pm 0.3$ | **39.7** $\pm 0.5$ | **42.1** $\pm 0.3$ |
| OpenBookQA (Mihaylov et al., 2018) | $29.8 \pm 0.6$ | **31.0** $\pm 0.8$ | **31.5** $\pm 0.4$ | **31.8** $\pm 0.8$ |
| PiQA (Bisk et al., 2020) | $65.4 \pm 0.6$ | **68.7** $\pm 0.3$ | **69.0** $\pm 0.5$ | **69.4** $\pm 0.5$ |
| Social IQA (Sap et al., 2019) | $41.7 \pm 0.3$ | $42.0 \pm 0.2$ | **42.7** $\pm 0.3$ | $42.6 \pm 0.7$ |
| WinoGrande (Sakaguchi et al., 2021) | $51.1 \pm 1.0$ | $51.2 \pm 0.4$ | $50.7 \pm 1.0$ | $50.9 \pm 0.4$ |
| MMLU (Hendrycks et al., 2021) | $28.6 \pm 0.2$ | $28.9 \pm 0.4$ | $28.5 \pm 0.2$ | $28.7 \pm 0.3$ |
| Average Performance | $39.5 \pm 0.3$ | $41.1 \pm 0.3$ | $41.2 \pm 0.3$ | $41.5 \pm 0.2$ |
| Beat Human on | – | 5 / 9 | 6 / 9 | 6 / 9 |
| Estimated FLOPs | 0 | 3.7e19 | 0 | 3.5e18 |

## C.4 THE EVALUATION RESULTS USING LIGHTEVAL

Following the approach of FineWeb (Penedo et al., 2024b), we employ the LightEval [6] library to evaluate our models using a suite of benchmarks selected for their stability and suitability. The chosen benchmarks exhibit three key characteristics: low score variance across different data samples, monotonic score improvement during training, and above-random baseline scores for models in the 1B parameter range. Table 8 presents the evaluation results. Our method, REGMIX, consistently

---

[6]https://github.com/huggingface/lighteval

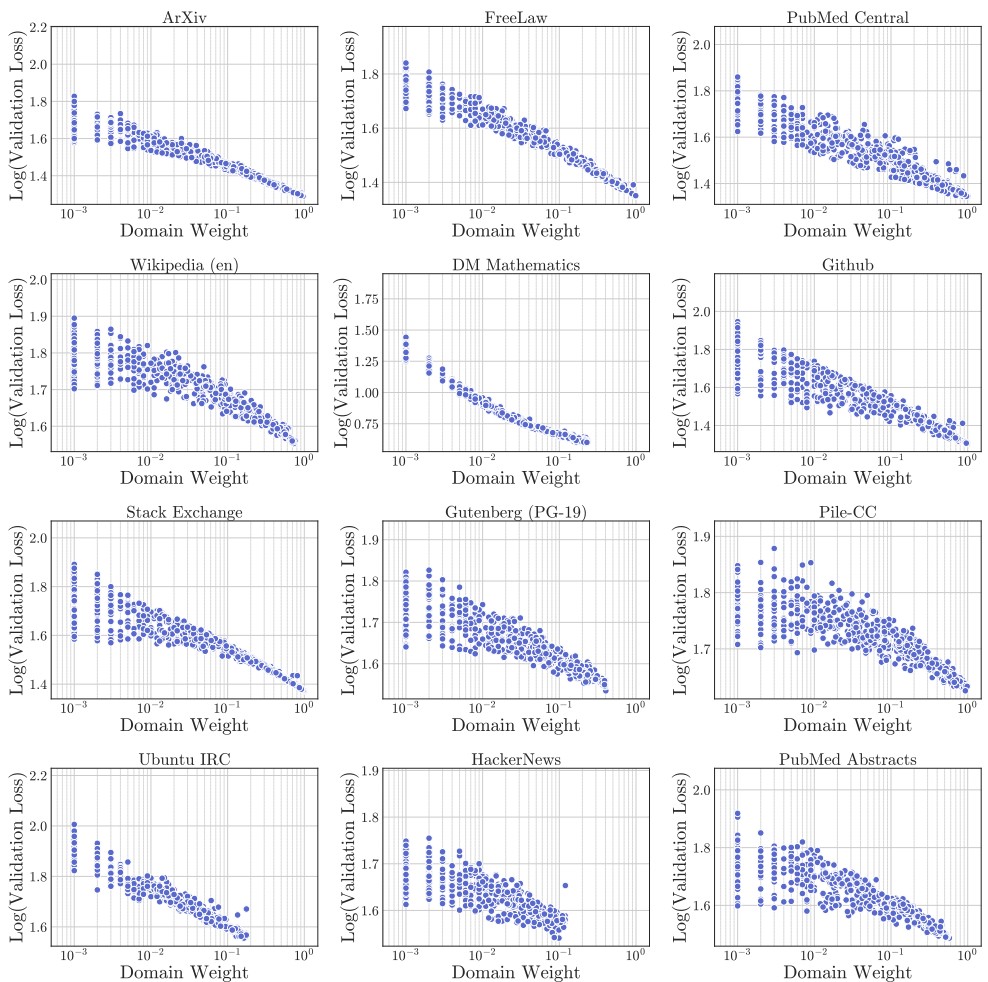

Figure 9: The visualization of 1M training logs across various data mixtures. The x-axis represents the weight of each domain in the data mixture and the y-axis shows the log value of validation loss for that domain. As seen, predicting the validation loss solely based on the domain weight is challenging.

outperforms the Human baseline on 6 benchmarks. Moreover, REGMIX demonstrates superior average performance compared to the DoReMi and the Pile-CC Only methods.

## C.5 DATA MIXTURE EFFECTS TRANSCEND SCALING LAWS

As shown in Figure 9, validation loss does not follow a simple log-log linear trend with domain weight, as scaling laws suggest. While DM Mathematics shows near-linear behavior, most domains, like Pile-CC, exhibit complex interactions that challenge straightforward predictions. This highlights the role of domain interactions in shaping loss, making individual weights insufficient for accurate modeling. Our approach addresses this by using the entire data mixture as input to a regression model, capturing cross-domain effects for better loss prediction.

# D   URL DOMAIN CORRELATION GRAPH

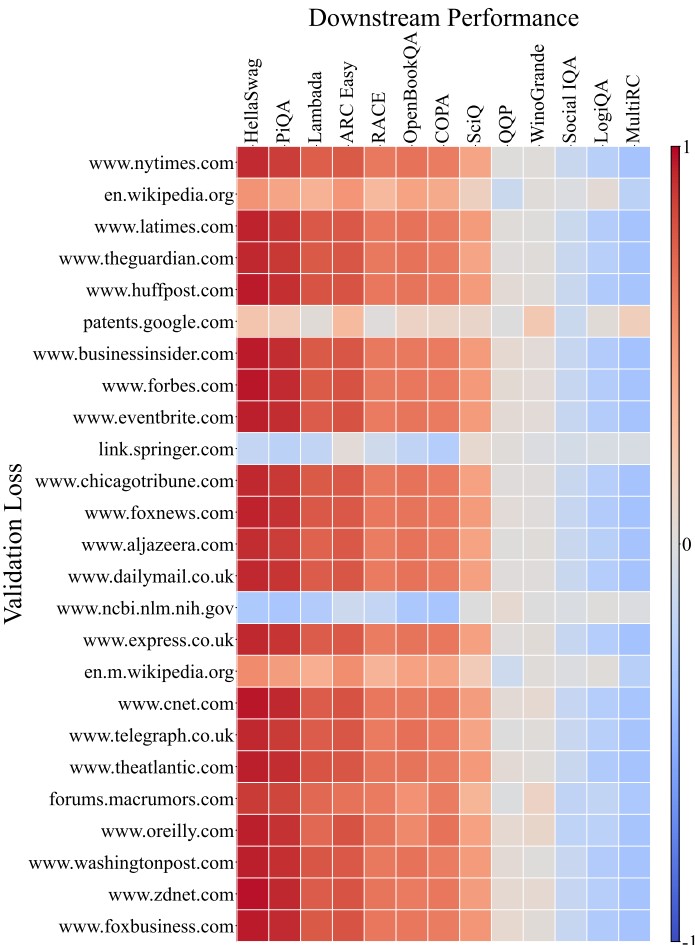

Figure 10: The visualization of correlations between different URL domains within the C4 subsets and the downstream performance (Part 1).

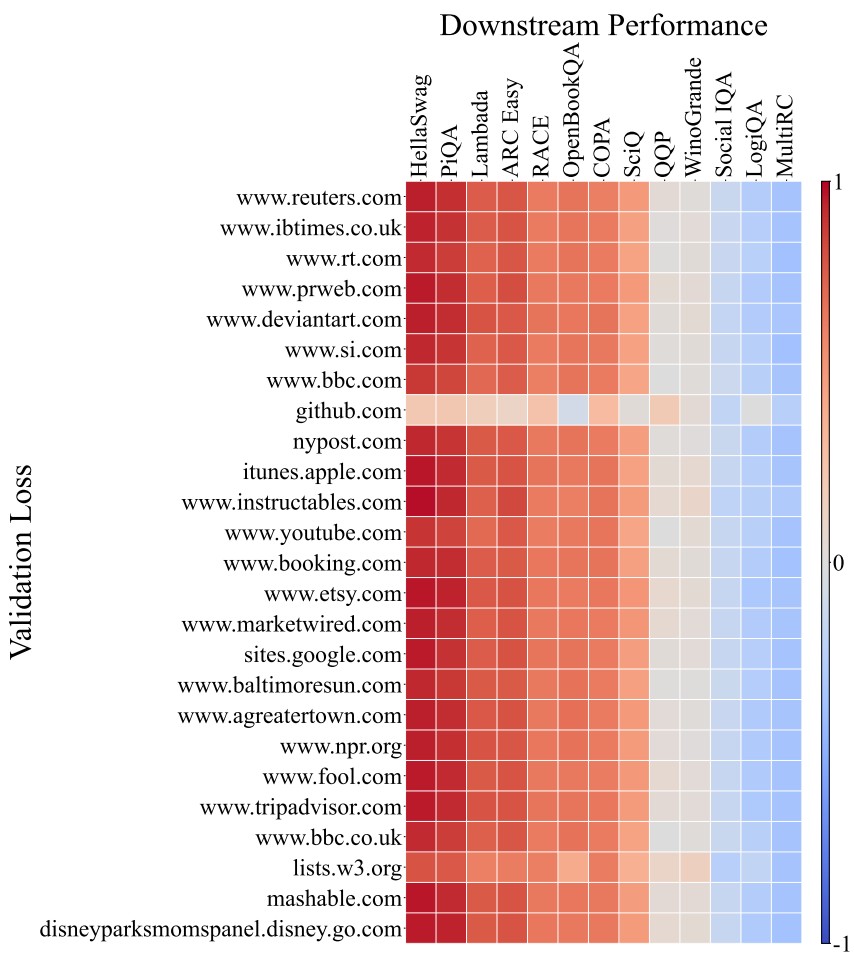

Figure 11: The visualization of correlations between different URL domains within the C4 subsets and the downstream performance (Part 2).

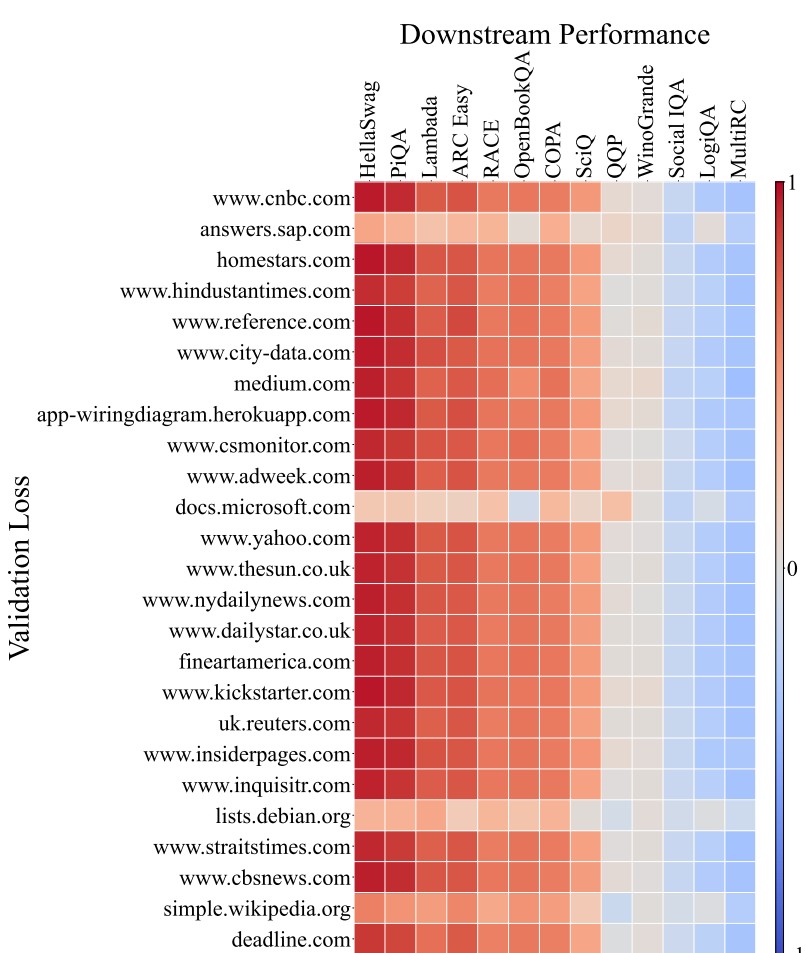

Figure 12: The visualization of correlations between different URL domains within the C4 subsets and the downstream performance (Part 3).

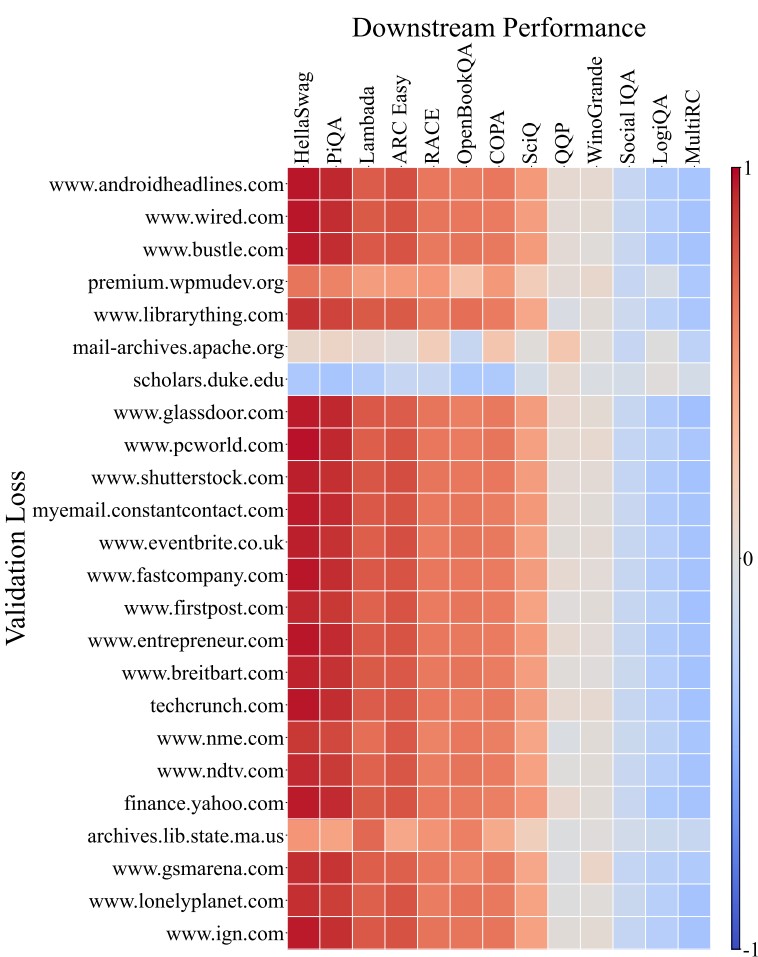

Figure 13: The visualization of correlations between different URL domains within the C4 subsets and the downstream performance (Part 4).

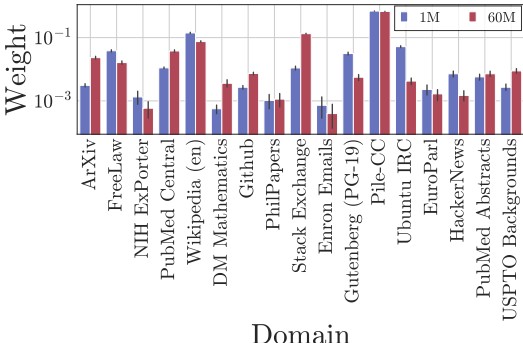

Figure 14: REGMIX yields similar data mixture distributions when using the 1M model and the 60M model as proxy models, demonstrating the stability of our method. Note that the y-axis is in log-scale for visualization purpose.

## E    IMPLEMENTATION DETAILS

We utilize the model architecture proposed by Zhang et al. (2024a) and create various model variants by modifying the number of layers, the number of attention heads, and the dimensions of token embeddings and hidden states, as illustrated in Figure 9. For tokenization, we employ the GPTNeoX tokenizer (Black et al., 2022), which has a vocabulary size of 50,432.

For models with 1M and 60M parameters, we set the training iterations as 1000 and the batch size as 1M tokens, which means the training budget is 1B tokens. Similarly, we train the larger model with 1B parameters with 25000 training iterations and the same batch size thus consuming 25B tokens in total. We set the learning rate as 4e-4 and use the cosine learning rate scheduler.

For linear regression, we employ 5-fold cross-validation with ridge regression to determine the optimal $\ell_2$ regularization weight from the set [1e-3, 1e-2, 1e-1, 1e0, 1e1, 1e2, 1e3]. For LightGBM, we manually set the number of iterations to 1000 and the learning rate to 1e-2. leaving all other hyperparameters at their default values.

Table 9: The detailed model configuration for different model sizes.

| Model | 1M | 60M | 1B | 7B |
|---|---|---|---|---|
| Vocabulary Size | 50432 | 50432 | 50432 | 50432 |
| $n_{\text{layers}}$ | 2 | 10 | 22 | 32 |
| $n_{\text{heads}}$ | 8 | 8 | 16 | 16 |
| $d_{\text{embedding}}$ | 256 | 768 | 2048 | 4096 |
| $d_{\text{model}}$ | 512 | 1536 | 5632 | 12288 |

## F    THE STABILITY OF OUR METHOD

Previous research (Xie et al., 2023a; Fan et al., 2023; Albalak et al., 2023) has employed small-scale proxy models, trained on substantial volumes of tokens, to predict optimal data mixtures for large language models. However, these approaches often suffer from instability issues. For example, DoReMi (Xie et al., 2023a) reported that different proxy model sizes can result in significantly different predicted data mixtures. Their findings (Figure 8, Appendix) show that using a 280M proxy model resulted in a Pile-CC weight of 0.67, while a 1B proxy model yielded a Pile-CC weight below 0.20. The large discrepancy highlights potential instabilities in previous approaches. To evaluate the robustness of REGMIX against such instabilities, we conducted comparative experiments using two distinct model scales: a 1M proxy model and a 60M proxy model. We used their respective training logs to fit regression models and subsequently simulated the top 1024 predictions. The resulting distributions are plotted in Figure 14. Our results demonstrate that while the prediction distributions for the 1M and 60M models are not identical, they exhibit remarkably similar patterns.

This consistency suggests that REGMIX achieves improved stability compared to previous approaches, even when varying the scale of proxy training models.

## G  DETAILED EXPERIMENTAL RESULTS

To facilitate future research, we share all the data mixtures and the corresponding downstream performances of the 64 trained models with 1B parameters.

| Model Index | 1 | 2 | 3 | 4 | 5 | 6 | 7 | 8 |
|---|---|---|---|---|---|---|---|---|
| *Pre-training Domain Weights* | | | | | | | | |
| ArXiv | 0.123 | 0.066 | 0.055 | 0.059 | 0.201 | 0.036 | 0.042 | 0.126 |
| FreeLaw | 0.065 | 0.071 | 0.052 | 0.083 | 0.004 | 0.212 | 0.113 | 0.21 |
| NIH ExPorter | 0.0 | 0.0 | 0.004 | 0.0 | 0.014 | 0.0 | 0.0 | 0.0 |
| PubMed Central | 0.126 | 0.211 | 0.177 | 0.174 | 0.243 | 0.153 | 0.089 | 0.123 |
| Wikipedia (en) | 0.036 | 0.013 | 0.02 | 0.177 | 0.01 | 0.005 | 0.022 | 0.055 |
| DM Mathematics | 0.0 | 0.0 | 0.011 | 0.0 | 0.03 | 0.047 | 0.007 | 0.008 |
| Github | 0.034 | 0.153 | 0.095 | 0.194 | 0.017 | 0.205 | 0.028 | 0.008 |
| PhilPapers | 0.0 | 0.033 | 0.0 | 0.0 | 0.0 | 0.0 | 0.0 | 0.0 |
| Stack Exchange | 0.039 | 0.097 | 0.18 | 0.0 | 0.103 | 0.075 | 0.011 | 0.129 |
| Enron Emails | 0.0 | 0.0 | 0.0 | 0.0 | 0.0 | 0.0 | 0.0 | 0.0 |
| Gutenberg (PG-19) | 0.0 | 0.0 | 0.016 | 0.0 | 0.002 | 0.0 | 0.217 | 0.035 |
| Pile-CC | 0.27 | 0.101 | 0.381 | 0.192 | 0.359 | 0.209 | 0.232 | 0.288 |
| Ubuntu IRC | 0.0 | 0.0 | 0.001 | 0.005 | 0.0 | 0.0 | 0.08 | 0.0 |
| EuroParl | 0.0 | 0.0 | 0.0 | 0.109 | 0.0 | 0.001 | 0.117 | 0.0 |
| HackerNews | 0.0 | 0.011 | 0.005 | 0.0 | 0.0 | 0.0 | 0.018 | 0.0 |
| PubMed Abstracts | 0.0 | 0.136 | 0.0 | 0.005 | 0.014 | 0.002 | 0.011 | 0.016 |
| USPTO Backgrounds | 0.307 | 0.106 | 0.003 | 0.0 | 0.002 | 0.055 | 0.011 | 0.0 |
| *Downstream Performance (%)* | | | | | | | | |
| Social IQA | 33.27 | 33.33 | 33.62 | 33.53 | 33.49 | 33.56 | 33.62 | 33.55 |
| HellaSwag | 40.58 | 36.86 | 40.58 | 36.06 | 40.07 | 37.85 | 37.93 | 39.59 |
| PiQA | 67.29 | 65.14 | 67.97 | 64.66 | 67.03 | 65.36 | 66.0 | 66.55 |
| OpenBookQA | 28.63 | 27.87 | 29.33 | 29.1 | 29.23 | 28.33 | 29.13 | 28.73 |
| Lambada | 29.17 | 26.86 | 31.55 | 27.11 | 29.16 | 28.92 | 31.53 | 30.92 |
| SciQ | 80.68 | 79.98 | 81.05 | 80.8 | 82.4 | 79.88 | 78.67 | 79.7 |
| COPA | 70.5 | 63.83 | 69.17 | 65.0 | 67.5 | 66.0 | 66.67 | 68.67 |
| RACE | 29.47 | 30.0 | 32.11 | 28.82 | 31.13 | 30.06 | 29.9 | 30.75 |
| ARC Easy | 50.03 | 48.72 | 50.01 | 46.64 | 51.06 | 47.46 | 46.75 | 48.39 |
| LogiQA | 23.76 | 24.17 | 25.29 | 25.29 | 24.55 | 25.96 | 25.45 | 26.32 |
| QQP | 55.71 | 55.9 | 54.84 | 56.52 | 54.01 | 56.34 | 52.35 | 54.2 |
| WinoGrande | 51.54 | 51.59 | 51.39 | 50.91 | 53.13 | 52.26 | 51.26 | 51.45 |
| MultiRC | 52.65 | 53.39 | 51.89 | 50.92 | 49.03 | 53.09 | 53.64 | 50.23 |
| **Avg** | 47.18 | 45.97 | 47.60 | 45.80 | 47.06 | 46.54 | 46.38 | 46.85 |

## H  PESUDOCODE OF REGMIX

To improve the clarity, we provide additional details regarding the regression model fitting process and the data used. Specifically, the regression model is trained using $N$ data points generated by evaluating proxy models on randomly sampled data mixtures. A pseudocode representation of the Algorithm 1 is included below to outline the procedure.

| Model Index | 9 | 10 | 11 | 12 | 13 | 14 | 15 | 16 |
|---|---|---|---|---|---|---|---|---|
| *Pre-training Domain Weights* | | | | | | | | |
| ArXiv | 0.184 | 0.226 | 0.107 | 0.139 | 0.101 | 0.099 | 0.251 | 0.147 |
| FreeLaw | 0.009 | 0.046 | 0.276 | 0.048 | 0.047 | 0.002 | 0.024 | 0.046 |
| NIH ExPorter | 0.0 | 0.0 | 0.0 | 0.0 | 0.001 | 0.022 | 0.0 | 0.0 |
| PubMed Central | 0.094 | 0.261 | 0.157 | 0.184 | 0.119 | 0.501 | 0.101 | 0.196 |
| Wikipedia (en) | 0.035 | 0.001 | 0.009 | 0.032 | 0.049 | 0.003 | 0.17 | 0.14 |
| DM Mathematics | 0.007 | 0.001 | 0.0 | 0.001 | 0.092 | 0.0 | 0.0 | 0.008 |
| Github | 0.106 | 0.189 | 0.024 | 0.055 | 0.078 | 0.017 | 0.048 | 0.237 |
| PhilPapers | 0.0 | 0.0 | 0.0 | 0.0 | 0.0 | 0.043 | 0.019 | 0.0 |
| Stack Exchange | 0.142 | 0.077 | 0.051 | 0.109 | 0.002 | 0.065 | 0.007 | 0.06 |
| Enron Emails | 0.0 | 0.0 | 0.0 | 0.0 | 0.0 | 0.0 | 0.0 | 0.0 |
| Gutenberg (PG-19) | 0.0 | 0.01 | 0.001 | 0.0 | 0.051 | 0.091 | 0.0 | 0.012 |
| Pile-CC | 0.341 | 0.114 | 0.273 | 0.354 | 0.283 | 0.055 | 0.339 | 0.111 |
| Ubuntu IRC | 0.0 | 0.003 | 0.0 | 0.0 | 0.057 | 0.0 | 0.017 | 0.0 |
| EuroParl | 0.0 | 0.0 | 0.003 | 0.003 | 0.0 | 0.006 | 0.0 | 0.0 |
| HackerNews | 0.002 | 0.0 | 0.034 | 0.0 | 0.0 | 0.0 | 0.0 | 0.001 |
| PubMed Abstracts | 0.005 | 0.039 | 0.009 | 0.075 | 0.061 | 0.007 | 0.0 | 0.01 |
| USPTO Backgrounds | 0.075 | 0.033 | 0.056 | 0.0 | 0.057 | 0.088 | 0.024 | 0.032 |
| *Downstream Performance (%)* | | | | | | | | |
| Social IQA | 33.43 | 33.21 | 33.31 | 33.17 | 33.28 | 32.43 | 33.57 | 33.7 |
| HellaSwag | 40.05 | 35.89 | 39.55 | 39.89 | 38.63 | 36.18 | 39.52 | 35.94 |
| PiQA | 66.6 | 64.74 | 66.29 | 66.27 | 66.9 | 64.05 | 66.7 | 64.51 |
| OpenBookQA | 28.87 | 26.6 | 29.33 | 28.73 | 29.4 | 27.87 | 29.67 | 27.83 |
| Lambada | 31.39 | 27.37 | 30.32 | 30.31 | 31.38 | 26.25 | 29.86 | 26.95 |
| SciQ | 81.1 | 79.12 | 79.97 | 82.85 | 79.42 | 81.4 | 81.38 | 81.23 |
| COPA | 67.0 | 64.5 | 66.83 | 69.5 | 67.33 | 65.83 | 69.5 | 66.33 |
| RACE | 30.57 | 29.63 | 30.49 | 30.85 | 30.35 | 28.66 | 31.21 | 29.57 |
| ARC Easy | 50.66 | 47.74 | 47.47 | 50.18 | 49.92 | 49.52 | 50.73 | 48.65 |
| LogiQA | 23.6 | 25.65 | 26.37 | 23.81 | 25.58 | 26.29 | 25.86 | 25.12 |
| QQP | 54.89 | 54.79 | 54.2 | 55.23 | 53.69 | 57.09 | 53.95 | 54.24 |
| WinoGrande | 50.83 | 51.84 | 51.05 | 51.83 | 52.12 | 52.0 | 51.01 | 51.82 |
| MultiRC | 54.18 | 54.48 | 50.17 | 52.12 | 51.42 | 52.69 | 51.87 | 53.48 |
| **Avg** | 47.17 | 45.81 | 46.57 | 47.29 | 46.88 | 46.17 | 47.30 | 46.11 |

---

**Algorithm 1** REGMIX: Data Mixture as Regression

---

1: **Input:** Token mixtures for $n$ domains $\mathbf{x}^0 = \{x_1^0, x_2^0, \ldots, x_n^0\}$, number of proxy models $N$, target metric $y$, and regression model $f$
2: **Output:** Optimal data mixture $\mathbf{x}^* = \{x_1^*, x_2^*, \ldots, x_n^*\}$
3: **Step 1: Train Proxy Models**
4: Generate $N$ random data mixtures $\{\mathbf{x}_i\}_{i=1}^N$, where each $\mathbf{x}_i = \{x_1^i, x_2^i, \ldots, x_n^i\}$ is sampled from a Dirichlet distribution $\text{Dir}(\alpha = \lambda \cdot \mathbf{x}^0)$, with $\lambda \in [0.1, 5.0]$ to ensure diversity.
5: **for** each mixture $\mathbf{x}_i$ **do**
6:     Train a small-scale proxy model using $\mathbf{x}_i$ for a fixed number of tokens.
7:     Evaluate the proxy model to compute the target metric $y_i$ (e.g., validation loss).
8: **end for**
9: **Step 2: Fit Regression Model**
10: Use $\{(\mathbf{x}_i, y_i)\}_{i=1}^N$ to train the regression model $f(\mathbf{x})$ to predict $y$.
11: **Step 3: Simulate Data Mixtures and Predict Performance**
12: Generate a large set of candidate mixtures $\{\mathbf{x}_j\}_{j=1}^M$.
13: Predict the target metric for each mixture: $y_j = f(\mathbf{x}_j)$.
14: Identify the mixture $\mathbf{x}^*$ that minimizes $y$: $\mathbf{x}^* = \arg\min_{\mathbf{x}_j} f(\mathbf{x}_j)$.
15: **Step 4: Train Large-Scale Model**
16: Use the identified optimal mixture $\mathbf{x}^*$ to train a large-scale model with significantly more tokens. Optionally, average top-performing mixtures for robustness.
17: **Return:** Optimal data mixture $\mathbf{x}^*$

---

| Model Index | 17 | 18 | 19 | 20 | 21 | 22 | 23 | 24 |
|---|---|---|---|---|---|---|---|---|
| *Pre-training Domain Weights* | | | | | | | | |
| ArXiv | 0.228 | 0.0 | 0.501 | 0.101 | 0.047 | 0.031 | 0.078 | 0.068 |
| FreeLaw | 0.016 | 0.019 | 0.005 | 0.03 | 0.014 | 0.073 | 0.024 | 0.181 |
| NIH ExPorter | 0.0 | 0.0 | 0.0 | 0.0 | 0.0 | 0.0 | 0.0 | 0.0 |
| PubMed Central | 0.204 | 0.084 | 0.156 | 0.272 | 0.163 | 0.053 | 0.302 | 0.126 |
| Wikipedia (en) | 0.02 | 0.159 | 0.17 | 0.021 | 0.218 | 0.129 | 0.027 | 0.07 |
| DM Mathematics | 0.036 | 0.009 | 0.0 | 0.099 | 0.0 | 0.0 | 0.0 | 0.001 |
| Github | 0.02 | 0.012 | 0.022 | 0.124 | 0.137 | 0.066 | 0.04 | 0.195 |
| PhilPapers | 0.004 | 0.0 | 0.017 | 0.0 | 0.0 | 0.0 | 0.0 | 0.0 |
| Stack Exchange | 0.002 | 0.052 | 0.062 | 0.113 | 0.173 | 0.12 | 0.007 | 0.24 |
| Enron Emails | 0.0 | 0.0 | 0.0 | 0.0 | 0.0 | 0.0 | 0.0 | 0.0 |
| Gutenberg (PG-19) | 0.0 | 0.001 | 0.002 | 0.054 | 0.001 | 0.089 | 0.002 | 0.0 |
| Pile-CC | 0.244 | 0.361 | 0.061 | 0.154 | 0.19 | 0.057 | 0.499 | 0.023 |
| Ubuntu IRC | 0.0 | 0.296 | 0.002 | 0.0 | 0.029 | 0.001 | 0.0 | 0.0 |
| EuroParl | 0.004 | 0.0 | 0.0 | 0.001 | 0.007 | 0.0 | 0.0 | 0.0 |
| HackerNews | 0.0 | 0.0 | 0.0 | 0.0 | 0.011 | 0.031 | 0.0 | 0.0 |
| PubMed Abstracts | 0.196 | 0.001 | 0.0 | 0.011 | 0.008 | 0.351 | 0.0 | 0.059 |
| USPTO Backgrounds | 0.026 | 0.007 | 0.002 | 0.02 | 0.001 | 0.001 | 0.021 | 0.036 |
| *Downstream Performance (%)* | | | | | | | | |
| Social IQA | 33.89 | 33.31 | 33.53 | 33.38 | 33.75 | 33.24 | 33.56 | 33.71 |
| HellaSwag | 38.68 | 39.9 | 34.67 | 37.12 | 37.44 | 36.07 | 42.15 | 34.67 |
| PiQA | 66.83 | 67.39 | 63.33 | 64.83 | 65.0 | 63.68 | 67.8 | 62.99 |
| OpenBookQA | 28.13 | 30.67 | 28.03 | 29.4 | 27.67 | 27.77 | 29.37 | 25.83 |
| Lambada | 28.78 | 28.56 | 24.13 | 29.41 | 27.67 | 28.03 | 33.47 | 24.04 |
| SciQ | 79.6 | 78.83 | 77.42 | 78.98 | 78.95 | 78.72 | 81.83 | 79.12 |
| COPA | 65.17 | 68.17 | 65.33 | 67.33 | 67.67 | 62.67 | 69.83 | 65.83 |
| RACE | 28.74 | 30.03 | 29.76 | 29.49 | 30.77 | 29.76 | 31.21 | 27.91 |
| ARC Easy | 48.86 | 49.42 | 47.9 | 48.3 | 47.88 | 46.68 | 50.92 | 45.24 |
| LogiQA | 25.91 | 26.34 | 26.24 | 25.76 | 26.11 | 26.24 | 24.17 | 25.91 |
| QQP | 53.35 | 53.18 | 50.61 | 51.49 | 54.27 | 54.99 | 52.77 | 55.19 |
| WinoGrande | 52.54 | 51.17 | 52.01 | 51.09 | 52.13 | 52.03 | 52.5 | 50.28 |
| MultiRC | 51.49 | 52.45 | 55.4 | 54.87 | 51.73 | 49.49 | 50.61 | 50.29 |
| **Avg** | 46.30 | 46.88 | 45.26 | 46.27 | 46.23 | 45.34 | 47.71 | 44.69 |

| Model Index | 25 | 26 | 27 | 28 | 29 | 30 | 31 | 32 |
|---|---|---|---|---|---|---|---|---|
| *Pre-training Domain Weights* | | | | | | | | |
| ArXiv | 0.074 | 0.076 | 0.05 | 0.067 | 0.244 | 0.073 | 0.234 | 0.08 |
| FreeLaw | 0.214 | 0.085 | 0.039 | 0.052 | 0.023 | 0.087 | 0.015 | 0.134 |
| NIH ExPorter | 0.0 | 0.0 | 0.0 | 0.0 | 0.0 | 0.026 | 0.0 | 0.0 |
| PubMed Central | 0.135 | 0.214 | 0.049 | 0.221 | 0.064 | 0.175 | 0.086 | 0.255 |
| Wikipedia (en) | 0.011 | 0.005 | 0.068 | 0.052 | 0.151 | 0.017 | 0.287 | 0.058 |
| DM Mathematics | 0.0 | 0.0 | 0.019 | 0.0 | 0.0 | 0.101 | 0.026 | 0.037 |
| Github | 0.121 | 0.127 | 0.042 | 0.101 | 0.073 | 0.1 | 0.04 | 0.171 |
| PhilPapers | 0.006 | 0.0 | 0.0 | 0.0 | 0.0 | 0.019 | 0.0 | 0.0 |
| Stack Exchange | 0.024 | 0.204 | 0.146 | 0.001 | 0.02 | 0.054 | 0.022 | 0.015 |
| Enron Emails | 0.0 | 0.0 | 0.0 | 0.0 | 0.0 | 0.0 | 0.0 | 0.0 |
| Gutenberg (PG-19) | 0.001 | 0.147 | 0.01 | 0.265 | 0.017 | 0.0 | 0.0 | 0.045 |
| Pile-CC | 0.088 | 0.138 | 0.302 | 0.214 | 0.383 | 0.12 | 0.134 | 0.182 |
| Ubuntu IRC | 0.001 | 0.002 | 0.0 | 0.026 | 0.01 | 0.134 | 0.0 | 0.0 |
| EuroParl | 0.0 | 0.0 | 0.008 | 0.0 | 0.0 | 0.037 | 0.0 | 0.0 |
| HackerNews | 0.004 | 0.0 | 0.0 | 0.0 | 0.0 | 0.0 | 0.0 | 0.0 |
| PubMed Abstracts | 0.132 | 0.001 | 0.01 | 0.002 | 0.007 | 0.053 | 0.022 | 0.016 |
| USPTO Backgrounds | 0.189 | 0.001 | 0.255 | 0.0 | 0.007 | 0.002 | 0.134 | 0.008 |
| *Downstream Performance (%)* | | | | | | | | |
| Social IQA | 33.51 | 33.4 | 33.59 | 33.52 | 33.53 | 33.49 | 33.16 | 33.56 |
| HellaSwag | 36.75 | 36.97 | 40.81 | 38.25 | 40.28 | 35.71 | 37.37 | 37.39 |
| PiQA | 64.09 | 64.74 | 67.97 | 66.15 | 66.88 | 63.84 | 64.47 | 65.05 |
| OpenBookQA | 29.47 | 28.7 | 29.57 | 29.77 | 29.5 | 29.13 | 29.47 | 28.0 |
| Lambada | 26.69 | 33.0 | 31.6 | 33.08 | 31.49 | 27.69 | 26.99 | 29.54 |
| SciQ | 80.03 | 79.17 | 80.12 | 80.22 | 81.92 | 78.23 | 77.42 | 80.87 |
| COPA | 67.67 | 65.5 | 69.0 | 65.67 | 68.33 | 63.33 | 64.67 | 67.17 |
| RACE | 30.05 | 30.19 | 30.96 | 30.37 | 30.08 | 29.62 | 30.13 | 29.92 |
| ARC Easy | 47.5 | 46.9 | 50.26 | 48.57 | 50.55 | 46.96 | 48.77 | 48.79 |
| LogiQA | 27.24 | 25.55 | 25.86 | 24.37 | 25.32 | 25.12 | 26.4 | 24.3 |
| QQP | 49.68 | 55.43 | 50.94 | 50.91 | 51.99 | 53.53 | 49.53 | 51.36 |
| WinoGrande | 51.68 | 52.12 | 51.93 | 51.5 | 52.32 | 51.67 | 52.13 | 52.63 |
| MultiRC | 51.24 | 51.91 | 50.33 | 52.42 | 52.52 | 54.04 | 52.05 | 53.04 |
| **Avg** | 45.82 | 46.43 | 47.15 | 46.52 | 47.29 | 45.57 | 45.58 | 46.28 |

| Model Index | 33 | 34 | 35 | 36 | 37 | 38 | 39 | 40 |
|---|---|---|---|---|---|---|---|---|
| *Pre-training Domain Weights* | | | | | | | | |
| ArXiv | 0.105 | 0.295 | 0.142 | 0.279 | 0.052 | 0.251 | 0.239 | 0.157 |
| FreeLaw | 0.007 | 0.029 | 0.122 | 0.01 | 0.07 | 0.007 | 0.087 | 0.062 |
| NIH ExPorter | 0.0 | 0.0 | 0.001 | 0.0 | 0.253 | 0.007 | 0.0 | 0.0 |
| PubMed Central | 0.407 | 0.061 | 0.065 | 0.184 | 0.4 | 0.331 | 0.223 | 0.039 |
| Wikipedia (en) | 0.045 | 0.124 | 0.0 | 0.0 | 0.003 | 0.107 | 0.029 | 0.096 |
| DM Mathematics | 0.054 | 0.0 | 0.001 | 0.0 | 0.0 | 0.0 | 0.0 | 0.007 |
| Github | 0.017 | 0.006 | 0.006 | 0.108 | 0.033 | 0.13 | 0.049 | 0.057 |
| PhilPapers | 0.0 | 0.0 | 0.003 | 0.0 | 0.0 | 0.0 | 0.0 | 0.0 |
| Stack Exchange | 0.126 | 0.006 | 0.001 | 0.097 | 0.019 | 0.021 | 0.202 | 0.174 |
| Enron Emails | 0.0 | 0.0 | 0.0 | 0.0 | 0.0 | 0.0 | 0.0 | 0.0 |
| Gutenberg (PG-19) | 0.009 | 0.047 | 0.014 | 0.039 | 0.0 | 0.001 | 0.0 | 0.015 |
| Pile-CC | 0.167 | 0.364 | 0.618 | 0.198 | 0.031 | 0.006 | 0.156 | 0.181 |
| Ubuntu IRC | 0.0 | 0.0 | 0.001 | 0.0 | 0.0 | 0.12 | 0.0 | 0.0 |
| EuroParl | 0.007 | 0.026 | 0.0 | 0.0 | 0.0 | 0.0 | 0.0 | 0.089 |
| HackerNews | 0.0 | 0.004 | 0.0 | 0.0 | 0.018 | 0.0 | 0.0 | 0.012 |
| PubMed Abstracts | 0.047 | 0.0 | 0.0 | 0.083 | 0.002 | 0.005 | 0.012 | 0.016 |
| USPTO Backgrounds | 0.008 | 0.037 | 0.025 | 0.002 | 0.119 | 0.014 | 0.001 | 0.095 |
| *Downstream Performance (%)* | | | | | | | | |
| Social IQA | 33.48 | 33.28 | 33.35 | 33.29 | 33.63 | 33.61 | 33.21 | 33.61 |
| HellaSwag | 38.0 | 40.18 | 43.37 | 37.69 | 32.96 | 32.98 | 37.31 | 37.79 |
| PiQA | 65.3 | 66.68 | 69.04 | 66.46 | 62.25 | 60.17 | 65.24 | 65.32 |
| OpenBookQA | 29.43 | 30.37 | 30.43 | 27.63 | 26.43 | 26.83 | 27.97 | 28.7 |
| Lambada | 26.59 | 31.46 | 31.71 | 30.21 | 18.92 | 20.29 | 28.1 | 28.58 |
| SciQ | 79.82 | 80.58 | 82.13 | 80.83 | 76.73 | 77.9 | 79.12 | 79.6 |
| COPA | 64.33 | 69.33 | 67.0 | 67.83 | 61.5 | 62.67 | 64.67 | 66.0 |
| RACE | 30.03 | 30.16 | 32.47 | 30.49 | 29.27 | 28.12 | 30.11 | 30.21 |
| ARC Easy | 48.86 | 49.88 | 52.22 | 48.32 | 44.86 | 45.54 | 48.15 | 48.86 |
| LogiQA | 25.91 | 24.3 | 23.35 | 24.96 | 26.19 | 27.68 | 25.47 | 25.37 |
| QQP | 56.06 | 56.56 | 52.57 | 56.7 | 52.54 | 48.04 | 49.81 | 57.12 |
| WinoGrande | 50.92 | 50.97 | 52.39 | 52.7 | 52.3 | 51.68 | 51.42 | 52.8 |
| MultiRC | 53.09 | 49.97 | 52.18 | 49.05 | 53.78 | 52.27 | 51.45 | 55.68 |
| **Avg** | 46.29 | 47.21 | 47.86 | 46.63 | 43.95 | 43.67 | 45.54 | 46.90 |

| Model Index | 41 | 42 | 43 | 44 | 45 | 46 | 47 | 48 |
|---|---|---|---|---|---|---|---|---|
| *Pre-training Domain Weights* | | | | | | | | |
| ArXiv | 0.422 | 0.466 | 0.027 | 0.063 | 0.121 | 0.041 | 0.033 | 0.114 |
| FreeLaw | 0.213 | 0.075 | 0.041 | 0.089 | 0.008 | 0.025 | 0.048 | 0.116 |
| NIH ExPorter | 0.0 | 0.0 | 0.0 | 0.0 | 0.0 | 0.0 | 0.0 | 0.0 |
| PubMed Central | 0.08 | 0.07 | 0.116 | 0.219 | 0.093 | 0.111 | 0.22 | 0.081 |
| Wikipedia (en) | 0.019 | 0.006 | 0.021 | 0.001 | 0.008 | 0.092 | 0.027 | 0.038 |
| DM Mathematics | 0.001 | 0.0 | 0.001 | 0.05 | 0.016 | 0.062 | 0.002 | 0.031 |
| Github | 0.026 | 0.044 | 0.067 | 0.291 | 0.012 | 0.121 | 0.169 | 0.109 |
| PhilPapers | 0.0 | 0.0 | 0.0 | 0.0 | 0.0 | 0.0 | 0.0 | 0.0 |
| Stack Exchange | 0.003 | 0.078 | 0.137 | 0.002 | 0.408 | 0.124 | 0.082 | 0.001 |
| Enron Emails | 0.0 | 0.0 | 0.0 | 0.0 | 0.0 | 0.0 | 0.0 | 0.0 |
| Gutenberg (PG-19) | 0.01 | 0.0 | 0.001 | 0.0 | 0.006 | 0.0 | 0.057 | 0.021 |
| Pile-CC | 0.026 | 0.2 | 0.549 | 0.238 | 0.156 | 0.214 | 0.312 | 0.428 |
| Ubuntu IRC | 0.0 | 0.0 | 0.002 | 0.0 | 0.013 | 0.129 | 0.0 | 0.001 |
| EuroParl | 0.0 | 0.0 | 0.0 | 0.001 | 0.001 | 0.006 | 0.0 | 0.0 |
| HackerNews | 0.0 | 0.0 | 0.0 | 0.001 | 0.0 | 0.012 | 0.0 | 0.0 |
| PubMed Abstracts | 0.101 | 0.028 | 0.002 | 0.045 | 0.005 | 0.012 | 0.0 | 0.031 |
| USPTO Backgrounds | 0.099 | 0.031 | 0.037 | 0.0 | 0.153 | 0.052 | 0.05 | 0.029 |
| *Downstream Performance (%)* | | | | | | | | |
| Social IQA | 33.49 | 33.43 | 33.07 | 33.28 | 33.44 | 33.08 | 33.78 | 33.17 |
| HellaSwag | 34.51 | 37.59 | 42.69 | 37.37 | 38.31 | 38.3 | 39.67 | 41.07 |
| PiQA | 62.24 | 65.58 | 68.05 | 66.62 | 66.54 | 65.52 | 66.98 | 67.21 |
| OpenBookQA | 27.1 | 28.77 | 28.9 | 28.07 | 28.07 | 27.6 | 31.17 | 29.73 |
| Lambada | 22.78 | 26.99 | 31.34 | 29.51 | 27.87 | 29.47 | 30.34 | 32.71 |
| SciQ | 77.78 | 80.25 | 79.47 | 80.25 | 80.7 | 79.72 | 81.35 | 81.77 |
| COPA | 64.0 | 66.33 | 67.0 | 67.0 | 67.33 | 68.33 | 67.17 | 67.67 |
| RACE | 28.33 | 28.82 | 30.78 | 30.8 | 30.08 | 30.24 | 30.24 | 30.67 |
| ARC Easy | 45.48 | 48.64 | 51.49 | 46.99 | 48.79 | 48.05 | 49.58 | 49.49 |
| LogiQA | 24.83 | 24.96 | 24.76 | 23.25 | 26.06 | 25.55 | 24.32 | 24.68 |
| QQP | 50.27 | 54.73 | 53.96 | 57.0 | 53.73 | 51.19 | 57.52 | 56.91 |
| WinoGrande | 51.79 | 51.63 | 51.32 | 50.76 | 53.18 | 52.45 | 50.72 | 52.24 |
| MultiRC | 54.03 | 53.96 | 48.91 | 50.74 | 53.01 | 50.89 | 47.63 | 53.84 |
| **Avg** | 44.35 | 46.28 | 47.06 | 46.28 | 46.7 | 46.18 | 46.96 | 47.78 |

| Model Index | 49 | 50 | 51 | 52 | 53 | 54 | 55 | 56 |
|---|---|---|---|---|---|---|---|---|
| *Pre-training Domain Weights* | | | | | | | | |
| ArXiv | 0.082 | 0.091 | 0.194 | 0.011 | 0.039 | 0.294 | 0.012 | 0.25 |
| FreeLaw | 0.12 | 0.084 | 0.04 | 0.022 | 0.063 | 0.119 | 0.16 | 0.058 |
| NIH ExPorter | 0.0 | 0.0 | 0.022 | 0.0 | 0.0 | 0.0 | 0.0 | 0.0 |
| PubMed Central | 0.051 | 0.343 | 0.126 | 0.37 | 0.079 | 0.186 | 0.311 | 0.104 |
| Wikipedia (en) | 0.067 | 0.0 | 0.046 | 0.006 | 0.0 | 0.023 | 0.014 | 0.044 |
| DM Mathematics | 0.034 | 0.174 | 0.028 | 0.0 | 0.002 | 0.005 | 0.0 | 0.0 |
| Github | 0.205 | 0.144 | 0.048 | 0.14 | 0.482 | 0.023 | 0.117 | 0.028 |
| PhilPapers | 0.0 | 0.0 | 0.01 | 0.0 | 0.0 | 0.0 | 0.0 | 0.0 |
| Stack Exchange | 0.036 | 0.009 | 0.099 | 0.058 | 0.012 | 0.001 | 0.004 | 0.06 |
| Enron Emails | 0.0 | 0.0 | 0.0 | 0.0 | 0.0 | 0.0 | 0.0 | 0.0 |
| Gutenberg (PG-19) | 0.0 | 0.019 | 0.04 | 0.216 | 0.0 | 0.002 | 0.236 | 0.0 |
| Pile-CC | 0.371 | 0.122 | 0.229 | 0.101 | 0.269 | 0.213 | 0.037 | 0.363 |
| Ubuntu IRC | 0.0 | 0.001 | 0.0 | 0.033 | 0.0 | 0.023 | 0.007 | 0.0 |
| EuroParl | 0.0 | 0.003 | 0.002 | 0.0 | 0.0 | 0.0 | 0.0 | 0.0 |
| HackerNews | 0.0 | 0.001 | 0.0 | 0.002 | 0.0 | 0.0 | 0.0 | 0.0 |
| PubMed Abstracts | 0.029 | 0.006 | 0.089 | 0.026 | 0.002 | 0.024 | 0.007 | 0.086 |
| USPTO Backgrounds | 0.004 | 0.004 | 0.027 | 0.015 | 0.052 | 0.088 | 0.094 | 0.007 |
| *Downstream Performance (%)* | | | | | | | | |
| Social IQA | 33.53 | 33.74 | 33.37 | 33.41 | 32.96 | 33.88 | 33.75 | 33.79 |
| HellaSwag | 39.09 | 35.65 | 38.68 | 36.07 | 37.68 | 38.53 | 35.4 | 40.5 |
| PiQA | 66.81 | 64.58 | 65.68 | 63.99 | 65.85 | 65.76 | 64.51 | 66.89 |
| OpenBookQA | 29.13 | 27.57 | 28.27 | 29.1 | 29.43 | 28.73 | 28.3 | 29.87 |
| Lambda | 30.23 | 26.19 | 30.29 | 30.84 | 29.76 | 29.03 | 28.63 | 30.74 |
| SciQ | 79.9 | 80.83 | 78.4 | 80.03 | 81.38 | 80.92 | 77.75 | 82.07 |
| COPA | 68.17 | 61.83 | 67.0 | 66.0 | 66.17 | 63.17 | 66.33 | 64.0 |
| RACE | 31.42 | 29.35 | 30.41 | 31.08 | 30.77 | 29.73 | 30.8 | 31.42 |
| ARC Easy | 49.54 | 47.71 | 49.02 | 47.64 | 48.38 | 49.36 | 46.96 | 51.22 |
| LogiQA | 24.99 | 24.58 | 25.32 | 24.91 | 25.17 | 26.22 | 24.63 | 24.91 |
| QQP | 54.06 | 56.48 | 50.96 | 56.62 | 56.45 | 53.86 | 53.85 | 53.26 |
| WinoGrande | 50.51 | 50.26 | 51.83 | 51.33 | 52.18 | 51.89 | 51.59 | 50.5 |
| MultiRC | 50.25 | 54.37 | 50.94 | 52.38 | 51.21 | 55.34 | 54.52 | 50.5 |
| **Avg** | 46.74 | 45.63 | 46.17 | 46.42 | 46.72 | 46.65 | 45.92 | 46.90 |

| Model Index | 57 | 58 | 59 | 60 | 61 | 62 | 63 | 64 |
|---|---|---|---|---|---|---|---|---|
| *Pre-training Domain Weights* | | | | | | | | |
| ArXiv | 0.137 | 0.176 | 0.471 | 0.081 | 0.107 | 0.278 | 0.119 | 0.131 |
| FreeLaw | 0.085 | 0.007 | 0.038 | 0.153 | 0.016 | 0.141 | 0.085 | 0.006 |
| NIH ExPorter | 0.0 | 0.0 | 0.0 | 0.0 | 0.0 | 0.0 | 0.027 | 0.03 |
| PubMed Central | 0.085 | 0.05 | 0.218 | 0.17 | 0.218 | 0.257 | 0.294 | 0.075 |
| Wikipedia (en) | 0.059 | 0.122 | 0.005 | 0.017 | 0.003 | 0.099 | 0.02 | 0.0 |
| DM Mathematics | 0.0 | 0.001 | 0.0 | 0.033 | 0.0 | 0.009 | 0.073 | 0.093 |
| Github | 0.039 | 0.088 | 0.097 | 0.041 | 0.238 | 0.041 | 0.038 | 0.369 |
| PhilPapers | 0.0 | 0.069 | 0.0 | 0.048 | 0.0 | 0.0 | 0.0 | 0.0 |
| Stack Exchange | 0.017 | 0.05 | 0.016 | 0.077 | 0.113 | 0.027 | 0.046 | 0.06 |
| Enron Emails | 0.009 | 0.0 | 0.0 | 0.0 | 0.0 | 0.0 | 0.001 | 0.0 |
| Gutenberg (PG-19) | 0.007 | 0.0 | 0.018 | 0.001 | 0.0 | 0.0 | 0.026 | 0.002 |
| Pile-CC | 0.435 | 0.339 | 0.112 | 0.268 | 0.272 | 0.128 | 0.232 | 0.188 |
| Ubuntu IRC | 0.0 | 0.006 | 0.017 | 0.095 | 0.001 | 0.0 | 0.0 | 0.001 |
| EuroParl | 0.0 | 0.012 | 0.0 | 0.0 | 0.0 | 0.0 | 0.001 | 0.003 |
| HackerNews | 0.001 | 0.0 | 0.0 | 0.0 | 0.0 | 0.0 | 0.0 | 0.017 |
| PubMed Abstracts | 0.004 | 0.004 | 0.001 | 0.0 | 0.02 | 0.0 | 0.013 | 0.016 |
| USPTO Backgrounds | 0.122 | 0.077 | 0.006 | 0.016 | 0.013 | 0.02 | 0.025 | 0.009 |
| *Downstream Performance (%)* | | | | | | | | |
| Social IQA | 33.24 | 33.3 | 33.56 | 33.54 | 33.42 | 33.84 | 33.32 | 33.55 |
| HellaSwag | 41.74 | 39.63 | 35.36 | 38.83 | 38.53 | 36.46 | 38.8 | 36.43 |
| PiQA | 68.07 | 67.31 | 64.44 | 66.38 | 66.5 | 64.74 | 66.54 | 64.87 |
| OpenBookQA | 29.2 | 29.5 | 28.1 | 27.97 | 27.83 | 27.37 | 28.83 | 27.87 |
| Lambada | 31.79 | 31.11 | 27.32 | 30.17 | 28.75 | 26.22 | 30.38 | 26.25 |
| SciQ | 80.42 | 79.83 | 80.85 | 79.6 | 78.93 | 80.05 | 79.5 | 78.65 |
| COPA | 66.17 | 69.0 | 64.0 | 64.83 | 67.0 | 64.0 | 66.0 | 66.83 |
| RACE | 31.39 | 29.82 | 29.67 | 30.08 | 29.98 | 29.46 | 30.37 | 29.19 |
| ARC Easy | 51.14 | 49.24 | 47.13 | 47.88 | 48.2 | 47.09 | 49.09 | 46.9 |
| LogiQA | 25.19 | 25.93 | 23.68 | 25.17 | 25.7 | 25.52 | 26.5 | 26.65 |
| QQP | 55.37 | 54.46 | 52.73 | 53.17 | 59.65 | 58.15 | 57.5 | 55.31 |
| WinoGrande | 53.21 | 51.46 | 50.83 | 52.16 | 52.37 | 51.41 | 51.63 | 51.85 |
| MultiRC | 53.58 | 52.31 | 52.22 | 53.03 | 50.41 | 52.17 | 52.27 | 51.5 |
| **Avg** | 47.73 | 47.15 | 45.38 | 46.37 | 46.71 | 45.88 | 46.98 | 45.84 |

# I    RANK INVARIANCE HYPOTHESIS

The rank invariance hypothesis asserts that the relative rankings of data mixtures should remain stable across varying model sizes (e.g., from 1M to 1B parameters) and token scales (e.g., from 1B to 25B tokens). This hypothesis suggests that the comparative effectiveness of different data mixtures does not significantly change as models grow in size or are trained on larger amounts of tokens.

To examine this hypothesis, we conducted a comprehensive set of experiments involving models of four distinct scales: 1M, 60M, 280M, and 1B parameters. Each model was trained using 64 different data mixture configurations, where the token amounts varied systematically. For each setting, we measured the validation loss for all data mixtures and determined their rankings. To quantify the consistency of rankings across model sizes and token scales, we computed the Spearman rank correlation coefficient ($\rho$) for each comparison.

The results shown in Figure 15, reveal high rank correlation coefficients across all model and token scales. This finding provides strong empirical support for the rank invariance hypothesis, indicating that the relative utility of data mixtures is robust to changes in model size and training tokens.

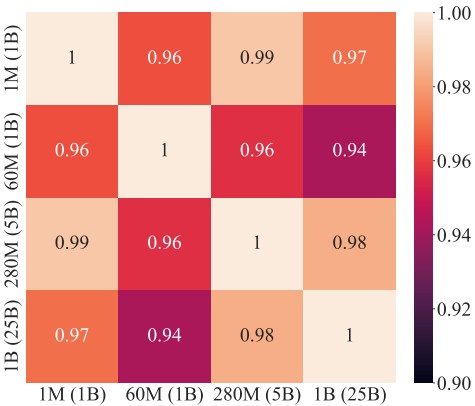

Figure 15: Heatmap of Spearman rank correlation coefficients ($\rho$) between validation loss rankings for different model scales (1M, 60M, 280M, and 1B parameters) and token scales (1B, 10B, and 25B tokens). The consistently high correlation values across all settings support the rank invariance hypothesis, indicating that the relative ranking of data mixtures remains stable despite changes in model size and training scale.

Table 10: Performance comparison between Human and RegMix on 7B models trained on 100B tokens across 13 downstream benchmarks.

| Benchmark | Human | REGMIX |
|---|---|---|
| Social IQA | 41.6 | 43.3 |
| HellaSwag | 55.0 | 63.3 |
| PiQA | 72.4 | 75.3 |
| OpenBookQA | 34.1 | 36.7 |
| Lambada | 44.9 | 51.0 |
| SciQ | 91.7 | 91.2 |
| ARC Easy | 63.5 | 65.4 |
| COPA | 75.0 | 80.1 |
| RACE | 35.1 | 36.6 |
| LogiQA | 25.7 | 24.1 |
| QQP | 58.9 | 56.1 |
| WinoGrande | 58.6 | 60.7 |
| MultiRC | 52.6 | 51.0 |
| Average | 54.5 | 56.5 (+2.0) |

## J    SCALING REGMIX TO 7B MODELS OVER 100B TOKENS

To further validate the effectiveness of our method on larger models, we conducted an experiment using Human baseline and our method data mixtures on a 7B model trained on 100B tokens. The results, summarized in Table I, demonstrate that RegMix still outperforms the Human baseline, achieving an average performance boost of 2%.

To provide a closer illustration, we benchmark the downstream performance of RegMix and Human on every dataset at intervals of 25B tokens in Figure 1. The results show that RegMix can significantly speed up pre-training, with a 50% acceleration on most benchmarks (e.g., HellaSwag) and up to 75% on some benchmarks (e.g., PiQA). Notably, the performance boost does not decrease with the amount of training tokens. However, we also observed that RegMix and Human struggle to improve on certain benchmarks (e.g., MultiRC), even with increased token amounts. These findings suggest that RegMix can almost benefit downstream tasks whose performance increases with the amount of training data, but may not improve tasks that do not follow scaling laws. This observation is quite intriguing and warrants further investigation.

## K    USING LARGER PROXY MODEL

We conducted a preliminary study on the impact of proxy model size on effectiveness. Specifically, we compared two configurations: (1) 128 proxy models of 1B parameters each, and (2) 512 proxy models of 1M parameters each (the setting used in our main experiments). Both configurations used 1B training tokens per proxy model. We limited our investigation to these configurations due to computational constraints that prevented us from exploring scenarios with more 1B-parameter proxy models.

Table 11: Performance comparison of optimized data mixtures derived from two proxy configurations ($512\times$ 1M and $128\times$ 1B) for 7B models trained on 100B tokens, evaluated across 13 downstream benchmarks.

| Benchmark | REGMIX (1B as proxy) | REGMIX (1M as proxy) |
|---|---|---|
| Social IQA | 43.4 | 43.3 |
| HellaSwag | 62.9 | 63.3 |
| PiQA | 75.1 | 75.3 |
| OpenBookQA | 36.2 | 36.7 |
| Lambada | 50.0 | 51.0 |
| SciQ | 91.2 | 91.2 |
| ARC Easy | 65.9 | 65.4 |
| COPA | 79.6 | 80.1 |
| RACE | 35.6 | 36.6 |
| LogiQA | 23.9 | 24.1 |
| QQP | 56.6 | 56.1 |
| WinoGrande | 60.7 | 60.7 |
| MultiRC | 51.6 | 51.0 |
| Average | 56.4 | 56.5 |

To evaluate these configurations, we used their respective optimized data mixtures to train two 7B models on 100B tokens and compared their performance. The results, summarized in Table K, show that both proxy settings achieved similar average performance across downstream tasks. This suggests that increasing proxy model size, even with fewer proxy models, can maintain competitive performance. However, given that the 1B proxy models do not significantly outperform the 1M proxy models, and considering that they incur over much more computational overhead, we recommend prioritizing a larger number of smaller proxy models over fewer larger ones. Based on our findings, we suggest practitioners begin with ultra-small proxy models (e.g., 1M parameters in our setting) as a starting point to optimize data mixtures for language model pre-training.

