# OpenReview forum: "RegMix: Data Mixture as Regression for Language Model Pre-training"
_ICLR.cc/2025/Conference — ICLR 2025 Spotlight_

### Official Review · Reviewer_uBLs · 2024-10-16

**Soundness:** 3
**Presentation:** 3
**Contribution:** 3
**Rating:** 8
**Confidence:** 3

**Summary:**

This paper proposes a method that trains a simple regression model over data mixture ratios and the LLM loss on very small models, and then use the trained regressor to predict the best data mixture configuration for training larger scale models. The method is interesting and the experiments are relatively comprehensive. The paper considers two regression methods and verified the predicted mixture on a 1B model.

**Strengths:**

1. the proposed method is novel and has practical impact to LLM training
2. the authors evaluated the predicted mixture on 1B model and compared to a few prior methods based on the downstream performance of the model
3. the paper also has good analysis and insights about regarding optimizing data mixture ratio, the relationship between validation PPL loss and downstream task performance, and the issues regarding scaling laws for data mixture.

**Weaknesses:**

1. there are some details of the regression model that's not clearly explained. It is not clear to me how the authors fit the regression model, how many data points are used to fit the models. It would be nice to have a pseudocode or open-sourced script
2. the mixture weights for the baseline DoReMi is directly taken from the paper. However, it's not clear if the optimal weights learned using the DoReMi method would be different due to model and data processing differences. It's probably better to re-learn the weights using the small model in the current experiment setup.

**Questions:**

1. in table 2, why do you only list the MSE for 1M model but rank correlation for all model sizes? what's the MSE for the other two sizes? Why is rank correlation a better metric here?
2. on line 243, the author mentioned the rank invariance hypothesis. However, it's not super clear to me what exactly this means and how this hypothesis is verified by the experimental results. Could you provide a clear definition of the rank invariance hypothesis and explicitly state how the experimental results support or verify this hypothesis?
3. there are some related work on data selection that falls into the group-level data selection category: https://aclanthology.org/2020.acl-main.754/

---

> ### Author Response · Authors · 2024-11-21
> **Rebuttal by Authors**
>
> > W1: there are some details of the regression model that's not clearly explained. It is not clear to me how the authors fit the regression model, how many data points are used to fit the models. It would be nice to have a pseudocode or open-sourced script
>
> Thank you for your suggestion. We have addressed this concern by including detailed pseudocode in Algorithm 1 to clarify the steps for fitting the regression model and the generation of data points. You can see the highlighted revision in $\textrm{\color{blue}Appendix I}$. Specifically, $N$ data points are created by training proxy models on diverse data mixtures sampled from a Dirichlet distribution. Each data point corresponds to a pair of mixture weights and the observed metric (e.g., validation loss). These data points are then used to train the regression model, which predicts the metric for unseen data mixtures. Additionally, we are committed to releasing the code, data, and models to ensure reproducibility.
>
> ---
>
> > W2: the mixture weights for the baseline DoReMi are directly taken from the paper. However, it's not clear if the optimal weights learned using the DoReMi method would be different due to model and data processing differences. It's probably better to re-learn the weights using the small model in the current experiment setup.
>
> We appreciate your suggestion regarding re-learning mixture weights and agree the proposed approach is more appropriate. While we made extensive efforts to reproduce the baseline methods across 17 domains using matching tokenizers and comparable proxy model sizes based on the released code, we encountered challenges in achieving consistently favorable results. The adapted data mixture from their paper, however, demonstrated strong experimental performance. To avoid potentially misleading comparisons, we decide to focus our current submission on presenting the adapted mixture results of baseline methods. Thank you for your suggestion!
>
> ---
>
> > Q1: in table 2, why do you only list the MSE for 1M model but rank correlation for all model sizes? what's the MSE for the other two sizes? Why is rank correlation a better metric here?
>
> Thanks for your detailed feedback! The MSE results of the 60M and 1B models are not included because MSE does not provide meaningful insights across different model sizes due to scale differences. Rank correlation ($\rho$), on the other hand, measures the relative order of performance across data mixtures and is invariant to scale, making it a better metric for cross-scale comparisons. Moreover, $\rho$ is more aligned with our assumption, the rank invariance hypothesis.
>
> To clarify, we have added a more detailed explanation in the revised submission, and please check out the updated caption in $\textrm{\color{blue}Table 2}$.
>
> ---
>
> > Q2: on line 243, the author mentioned the rank invariance hypothesis. However, it's not super clear to me what exactly this means and how this hypothesis is verified by the experimental results. Could you provide a clear definition of the rank invariance hypothesis and explicitly state how the experimental results support or verify this hypothesis?
>
> Thanks for your question! The rank invariance hypothesis posits that the ranking of data mixtures should remain stable across various model scales (e.g., from 1M parameters to 1B parameters) and token scales (e.g., from 1B tokens to 25B tokens).
>
> To clarify this concept further, we conduct additional experiments with models of 280M parameters, alongside models of 1M, 60M, and 1B parameters. Each of these four model scales was trained using the same 64 different data mixture configurations, varying in token amounts. We then compared the validation loss rankings across different data mixtures and calculated the rank correlation coefficient ($\rho$) for each setting. The correlation of each setting with the actual ranks of the 1B parameter models trained on 25B tokens is summarized below:
>
> | Setting | 1M (1B token) | 60M (1B token) | 280M (5B token) |
> | :---: | :---: | :---: | :---: |
> | 1B (25B token) | 96.97% | 94.22% | 98.43% |
>
> The above table demonstrates the strong correlation between the different models and token scales, supporting the rank invariance hypothesis. We have also added it to $\textrm{\color{blue}Appendix J}$ for full details of these experiments.
>
> ---
>
> > Q3: there are some related work on data selection that falls into the group-level data selection category: https://aclanthology.org/2020.acl-main.754/
>
> Thank you for pointing out the related work! We have cited this paper and placed it among the group-level data selection works in the revised submission. We find the method in the proposed paper very promising and see the potential for exploring multilingual balance optimization through a combination of RegMix and the approach. Thanks for pointing out!

---

### Official Review · Reviewer_wxJ7 · 2024-11-02

**Soundness:** 3
**Presentation:** 3
**Contribution:** 3
**Rating:** 6
**Confidence:** 4

**Summary:**

This paper formulates data mixing problems as a regression task. The authors propose to search data mixtures on small models and fit regression models to predict the optimal data mixture, which is then transferred to larger-scale model training. The authors empirically show the rankings of different data mixtures hold consistent between small and large-scale training. And data mixture found to be optimal at small scales can lead to improved performance compared to human heuristics and previous methods at large scales.

**Strengths:**

1. The paper is well-written and easy to follow.
2. The author identifies a helpful assumption, namely ranking invariance regarding training scales, which helps reduce the cost of tuning data mixtures in this paper.
3. The proposed method is simple to implement and efficient, thus appealing to try in practice.
4. The paper contains extensive experiments to show optimizing data mixtures improves model performance.

**Weaknesses:**

1. The feasibility of treating the data mixing problem as a regression problem has been an idea unveiled by previous studies [1,2], as the authors also mentioned in the paper.
2. The methods experiment on as many as 512 training runs. It is unclear whether the regression step is still necessary with so many experimented mixtures. This makes the proposed method actually a grid search with a small-scale proxy.
3. The proposed method highly depends on the assumption of ranking invariance regarding scales. The author only provides limited empirical results on this assumption. However, such an assumption is questionable according to [3]. It would be better if the authors provide more discussion to explain the scope where this assumption holds.

[1] Data Mixing Made Efficient: A Bivariate Scaling Law for Language Model Pretraining

[2] Data Mixing Laws: Optimizing Data Mixtures by Predicting Language Modeling Performance

[3] Scaling Laws for Data Filtering— Data Curation cannot be Compute Agnostic

**Questions:**

See weaknesses.

---

> ### Author Response · Authors · 2024-11-21
> **Rebuttal by Authors**
>
> > W1: The feasibility of treating the data mixing problem as a regression problem has been an idea unveiled by previous studies [1,2], as the authors also mentioned in the paper.
>
> Thank you for your comment. We very much like and respect the previous work like data mixing laws (DML), and please allow me to respectfully describe their differences. DML seeks analytical scaling functions to describe data mixing effects, but our RegMix approach directly optimizes target metrics through regression models. We believe regression models offer a more flexible and nuanced approach to understanding data mixture performance, allowing us to explore non-linear relationships and interactions that DML might be hard to model.
>
> ---
>
> > W2: The methods experiment on as many as 512 training runs. It is unclear whether the regression step is still necessary with so many experimented mixtures. This makes the proposed method actually a grid search with a small-scale proxy.
>
> Thank you for this important question. While 512 training runs may seem substantial, it's actually far more efficient than grid search, and here's why:
>
> Our work spans 17 domains from the Pile, which significantly increases the complexity of data mixture optimization. Even with a simplified approach of allocating 10 units of 0.1 across these 17 domains (maintaining a sum of 1), the computational demands remain considerable.
>
> This scenario represents a "stars and bars" problem in combinatorics - specifically, a combination problem with replacement. The total number of possible combinations can be calculated as C(n + k - 1, k), where n = 17 (domains) and k = 10 (units), yielding 5,311,735 combinations.
>
> While existing methods, like those in the Data Mixing Law paper, can use token availability to constrain the search space, the resulting space remains vast. Our approach, requiring only 512 runs, achieves feasible regression accuracy while reducing computational requirements by several orders of magnitude compared to traditional grid search, which would demand millions of training runs.
>
> Additionally, in response to the suggestion of Reviewer `YiAH`, we have extended RegMix to encompass 100 domains (please refer to our detailed response to Reviewer `YiAH` for more information). In this setting, we utilize 1,000 proxy models (1M parameters each) to achieve a correlation of 98.8% on the unseen data mixtures of 60M parameter models. We hope this addresses your concerns regarding the effectiveness of regression models using this challenging case.
>
> ---
>
> > W3: The proposed method highly depends on the assumption of ranking invariance regarding scales. The author only provides limited empirical results on this assumption.
> However, such an assumption is questionable according to [3].
> It would be better if the authors provided more discussion to explain the scope where this assumption holds.
>
> We completely agree with your point. As noted in $\textrm{\color{blue}Appendix B}$, many existing data mixing methods assume the availability of unlimited data for each domain, struggling with realistic scenarios. In contrast, RegMix can effectively manage token availability by adjusting the simulation space, particularly leveraging the 4-epoch practice introduced by Muennighoff et al. 2023 [1]. For instance, if we can afford to repeat HackerNews for 4 epochs and its token count constitutes 3% of the total expected training tokens, we can set its maximum domain weight to 12% during simulation. This approach allows RegMix to efficiently handle data mixture according to the available computational budget, in line with the findings of Goyal et al. 2024 [2]. Furthermore, exploring the integration of our method with the decay coefficient of data reuse proposed by Muennighoff et al. [1] could be an intriguing avenue for future research.
>
> [1] Muennighoff et al. "Scaling Data-Constrained Language Models", NeurIPS 2023.
>
> [2] Goyal et al. "Scaling Laws for Data Filtering--Data Curation cannot be Compute Agnostic." Proceedings of the IEEE/CVF Conference on Computer Vision and Pattern Recognition 2024.

---

> > ### Comment · Reviewer_wxJ7 · 2024-11-26
> >
> > Thank you for your detailed responses and extensive experiments.
> >
> > It is great to see the empirical justifications for the method's capability to handle as many as 1000 domains, which is much closer to real-world settings to my knowledge.
> >
> > The authors may misunderstand my concerns about the rank invariance assumption (W3) but I see similar concerns from other reviewers and the authors supplement experiments on more model scales. This can somehow support the authors' claim while I am still confused given the contradicting opinions from different literature. I tend to believe that rank invariance holds under some preconditions, which may hold true in most cases. Or the resolution of loss is too small to faithfully reflect the rankings, which requires us to make predictions on downstream tasks.
> >
> > Overall, I appreciate the authors' excellent work and would like to support its acceptance.

---

> > > ### Author Response · Authors · 2024-11-26
> > > **Thanks for your support**
> > >
> > > Thank you for sharing your thoughtful feedback on our work. We greatly appreciate your input for our work! We are particularly encouraged by your positive assessment of the scalability of RegMix with 1000 domains. As you note, this better reflects real-world deployment scenarios.
> > >
> > > While our experiments across model scales provide empirical support for the rank invariance hypothesis, we acknowledge there is still active discussion in the literature about it, and the conclusion is not decided yet. Your perspective that it may hold under certain preconditions is insightful. The suggestion about potential resolution limitations of losses is also well-taken.
> > >
> > > We will continue investigating the theoretical foundations and practical implications of RegMix and its hypothesis in future work. Thank you again for the constructive review that helped strengthen our paper.
> > >
> > > Best Regards,
> > >
> > > Authors

---

### Official Review · Reviewer_FGaq · 2024-11-02

**Soundness:** 2
**Presentation:** 3
**Contribution:** 2
**Rating:** 6
**Confidence:** 3

**Summary:**

The paper proposes a method called REGMIX for automatically selecting an effective data mixture to optimize the pre-training of large language models. REGMIX formulates the data mixture selection as a regression task, training a set of small models with diverse data mixtures and fitting a regression model to predict their performance. The fitted regression model is then used to simulate and identify the top-performing data mixture, which is subsequently used to train a large-scale model. The empirical results demonstrate that REGMIX can improve downstream task performance and achieves results comparable to or surpassing the DoReMi method while using only 10% of the compute budget.

**Strengths:**

- The paper presents a novel method, REGMIX, which formulates the data mixture selection problem as a regression task. This is a creative approach that leverages small-scale proxy models to predict optimal data mixtures for large-scale models.
- The authors conducted extensive experiments, training 512 models with 1M parameters on 1B tokens to fit the regression model. They then validated this model by training a 1B parameter model on 25B tokens, showing superior performance compared to human selection and the DoReMi method.
- The method allows for parallel training of small proxy models, making it more scalable than previous approaches that require training a single model for a long time.
- The paper provides several interesting findings, such as the significant impact of data mixtures on performance, the strong positive correlation of web corpora with downstream performance, and the complex interactions between domains.

**Weaknesses:**

- The key assumption of rank invariance of data mixtures across different model sizes and token counts is not thoroughly validated. This assumption might not hold in all cases, especially with significant changes in model scale and data distribution.
- The paper claims stability across different proxy model sizes, but the experiments are limited to models with up to 1B parameters. It remains unclear if the method would be equally effective for much larger models commonly used in practice (e.g., 7B or 70B parameters). If so, the additional computation cost could not be ignored.
- The authors only trained 25B tokens using the obtained data mixtures. This raises the question of whether the data scale could be enlarged to 50 times or even 100 times. And can LLM sitll benefit from the obtained data mixture?
- Although the method is more efficient than some previous approaches, training 512 small models still requires substantial computational resources. This could be a limitation for teams with limited access to such resources. The trade-off between performance gains and additional costs may not always hold when the model scales up.

**Questions:**

- Can the authors provide more theoretical or empirical evidence to support the rank invariance assumption? How does this assumption hold up with significant changes in model scale and data distribution?
- How does REGMIX perform with proxy models larger than 1B parameters? Can the authors provide any preliminary results or insights on this? Or could the obtained data mixtures guide us to train a better model using much more tokens, e.g., 100B?
- Can the authors provide a detailed comparison of the computational resources required by REGMIX and other methods? This would help in understanding the practical feasibility of the method.

---

> ### Author Response · Authors · 2024-11-21
> **Rebuttal by Authors [1/2]**
>
> > W1: The key assumption of rank invariance of data mixtures across different model sizes and token counts is not thoroughly validated. This assumption might not hold in all cases, especially with significant changes in model scale and data distribution.
>
> > Q1: Can the authors provide more theoretical or empirical evidence to support the rank invariance assumption? How does this assumption hold up with significant changes in model scale and data distribution?
>
> Thank you for your insightful questions regarding the rank invariance hypothesis. We acknowledge your concerns about the assumption of rank invariance across different model sizes and token counts. To address this, we conducted extensive experiments during the rebuttal period, incorporating models with 280M parameters as well as models with 1M, 60M, and 1B parameters. Each of these models was trained using the same 64 different data mixture configurations, allowing us to systematically analyze the ranking of validation losses.
>
> Our findings, summarized in the table below, illustrate the strong rank correlation ($\rho$) among different model and token scales:
>
> | Setting | 1M (1B token) | 60M (1B token) | 280M (5B token) |
> | :---: | :---: | :---: | :---: |
> | 1B (25B token) | 96.97% | 94.22% | 98.43% |
>
> These results indicate that even with significant changes in model scale (from 1M to 1B) and token counts (from 1B tokens to 25B tokens), the rankings of data mixtures remain remarkably stable, thereby supporting the rank invariance hypothesis. We have also updated $\textrm{\color{blue}Appendix J}$ to provide more details and the visualization.
>
> We appreciate your feedback and hope this additional evidence helps clarify the validity of our hypothesis.
>
> ---
>
> > W2: The paper claims stability across different proxy model sizes, but the experiments are limited to models with up to 1B parameters. It remains unclear if the method would be equally effective for much larger models commonly used in practice (e.g., 7B or 70B parameters). If so, the additional computation cost could not be ignored.
>
> > W3: The authors only trained 25B tokens using the obtained data mixtures. This raises the question of whether the data scale could be enlarged to 50 times or even 100 times. And can LLM still benefit from the obtained data mixture?
>
> We appreciate your concerns about the scalability of our method. To address your concerns about our current experimental setup, we propose the following additional pre-training experiments during the rebuttal period:
>
> * **Training a 7B model on 100B tokens** with our RegMix data mixture
> * **Training a 7B model on 100B tokens** with the Human baseline data mixture
>
> These proposed experiments will help validate the effectiveness of our data mixture approach at larger model scales and token counts. We welcome any suggestions to improve the experimental setup and provide more comprehensive evidence of our method's generalizability.
>
> Additionally, we want to clarify that our stability analysis focuses on 1M and 60M proxy models, both significantly smaller than 1B parameters. The overall computational cost of all 1M model training is **approximately 2% of 1B model** training over 25B tokens, which we consider negligible.
>
> ---
>
> > W4: Although the method is more efficient than some previous approaches, training 512 small models still requires substantial computational resources. This could be a limitation for teams with limited access to such resources. The trade-off between performance gains and additional costs may not always hold when the model scales up.
>
> We appreciate your concern about computational resources. Despite training 512 proxy models, **the total computational cost remains remarkably low due to our ultra-small model design**. Specifically, training 512x 1M models (with just 2 layers) consumes only approximately 2% of the computational resources required for training a 1B model. This means that researchers capable of training a 1B model on 25B tokens can comfortably accommodate our proposed methodology without significant additional computational burden.

---

> > ### Comment · Reviewer_FGaq · 2024-11-25
> > **After reading the rebuttal**
> >
> > Thank you for your efforts during the rebuttal phase. I appreciate how you addressed my concerns point by point. I have read your response as well as the opinions of the other reviewers. As I mentioned in my summary of strengths, I recognize the contributions made by the authors, however, I still have some reservations about how this method applies to the nature of training large language models. Based on your clarifications, I would like to raise my score accordingly.

---

> ### Author Response · Authors · 2024-11-21
> **Rebuttal by Authors [2/2]**
>
> > Q2: How does REGMIX perform with proxy models larger than 1B parameters? Can the authors provide any preliminary results or insights on this? Or could the obtained data mixtures guide us to train a better model using many more tokens, e.g., 100B?
>
> Thanks for your suggestion! We never thought of using 1B parameters as proxy model sizes, but we would love to conduct the experiments during the rebuttal to answer your question! We propose to explore 1B parameter proxy models by training 128x 1B models with 1B tokens (the most computation cost we can afford now) and comparing their optimized data mixture performance against our current best data mixture. Specifically, we will evaluate RegMix (1M) and RegMix (1B) in the context of training 7B models on 100B tokens during the rebuttal period, providing new insights into the scalability of our data mixture optimization approach across different model sizes. Please stay tuned for the results!
>
> > Q3: Can the authors provide a detailed comparison of the computational resources required by REGMIX and other methods? This would help in understanding the practical feasibility of the method.
>
> Thanks for your suggestion! To make it clear, we have previously reported the estimated FLOPs in $\textrm{\color{blue}Table 4}$, which indicates the extra computation cost of different methods. We will highlight the extra computation cost part in the final version. To make it clear for you, we copy the numbers of DoReMi (the baseline) and RegMix below:
>
> | Method | Average Performance | FLOPs |
> | :---: | :---: | :---: |
> | DoReMi | 46.8 | 3.7e19 |
> | RegMix | 47.3 | 3.5e18 (↓90%) |

---

> ### Author Response · Authors · 2024-11-25
> **Thank you for your feedback and we have 7B results now!**
>
> Thank you very much for your response and for adjusting your score accordingly, and we are also excited to share our latest findings! As mentioned previously, we have ongoing experiments with 7B models, and we are pleased to report that **RegMix still outperforms the Human baseline significantly on 7B models trained on 100B tokens**!
>
> To recap, in order to further validate the effectiveness of our method on larger models, we conducted an experiment using Human baseline and RegMix data mixtures on a 7B model trained on 100B tokens. The results, summarized in the following table ( ${\textrm{\color{blue}Table 10}}$ in the paper), demonstrate that RegMix still outperforms the Human baseline, achieving an average performance boost of 2%:
>
> | Benchmark | Human | Our Method |
> | --- | --- | --- |
> | Social IQA | 41.6 | 43.3 |
> | HellaSwag | 55.0 | 63.3 |
> | PiQA | 72.4 | 75.3 |
> | OpenBookQA | 34.1 | 36.7 |
> | Lambada | 44.9 | 51.0 |
> | SciQ | 91.7 | 91.2 |
> | ARC Easy | 63.5 | 65.4 |
> | COPA | 75.0 | 80.1 |
> | RACE | 35.1 | 36.6 |
> | LogiQA | 25.7 | 24.1 |
> | QQP | 58.9 | 56.1 |
> | WinoGrande | 58.6 | 60.7 |
> | MultiRC | 52.0 | 51.0 |
> | **Average** | **54.5** | **56.5 (+2.0)** |
>
> To provide a more detailed illustration, we benchmarked the downstream performance of RegMix and Human on every dataset at intervals of 25B tokens in $\textrm{\color{blue}Figure 16}$ of $\textrm{\color{blue}Appendix K}$. The results show that RegMix can significantly speed up pre-training, with a **50% acceleration on most benchmarks (e.g., HellaSwag)** and **up to 75% on some benchmarks (e.g., PiQA)**. Notably, the performance boost does not decrease with the amount of training tokens. However, we also observed that both RegMix and Human struggle to improve on certain benchmarks (e.g., MultiRC), even with increased token amounts. These findings suggest that **RegMix mainly benefits downstream tasks whose performance increases with the amount of training data, but may not improve tasks that do not follow scaling laws**. This observation is intriguing and worthy for further investigation.
>
> We would like to thank you for your encouraging feedback, which motivates us to explore whether RegMix works well on larger model sizes. Your feedback also raises an interesting insight: RegMix mainly benefits tasks that are aligned with scaling laws (i.e., performance increases with the amount of tokens). We hope these results will increase your confidence in the effectiveness of RegMix. Thank you again for your time and effort during the review!

---

### Official Review · Reviewer_MoX3 · 2024-11-04

**Soundness:** 3
**Presentation:** 4
**Contribution:** 4
**Rating:** 8
**Confidence:** 3

**Summary:**

The paper presents RegMix, a new method for optimizing data mixtures in pre-training large language models (LLMs). Recognizing the importance of data composition, the authors frame mixture selection as a regression task. RegMix uses small proxy models trained on various data mixtures to build a predictive model that identifies the optimal mixture for larger models.

The authors conduct extensive experiments to validate the approach, showing that models trained with RegMix-selected data mixtures outperform those trained with mixtures chosen by other methods, including human selection and DoReMi, while utilizing only 10% of the compute budget.

The authors provide insights into the effects of data mixtures, offering empirical evidence that data mixtures can significantly impact performance, with single-task performance variations reaching up to 14.6%. They also emphasize the complex interactions that occur between different data domains.

**Strengths:**

1. RegMix introduces a fresh approach by framing data mixture selection as a regression problem rather than relying on complex optimizations or heuristics, making the process scalable and computationally efficient.

2. The paper’s experimental setup is robust, with 512 small proxy models across diverse data mixtures, creating a solid regression model for data selection.

3. The paper is well-organized, clearly explaining the methodology and experiments. It introduces the hypothesis of rank invariance in data mixtures, supported by visual aids, making the regression model’s role easy to understand.

**Weaknesses:**

The paper conducts a set of small-proxy models trained with small-scale tokens.

The paper only experiments with 1M models with 1B tokens.
It is unclear how to decide the size of the proxy model parameter and training token.

**Questions:**

How do we decide the size for proxy model and training tokens.?

---

> ### Author Response · Authors · 2024-11-21
> **Rebuttal by Authors**
>
> > W1: The paper conducts a set of small-proxy models trained with small-scale tokens.
>
> Thank you for the observation! The decision to employ small-proxy models alongside small-scale tokens is primarily driven by a desire to maintain a balance between experimental feasibility and computational efficiency. Proxy models enable us to efficiently explore a diverse array of data mixtures, yielding predictions that exhibit strong generalizability to larger-scale models, as verified by our experiments.
>
> Although our approach encompasses a suite of small proxy models, the computation overhead is relatively small since our method leverages **ultra-small proxy models** (for instance, the 1M model). Therefore, the computational overhead attributed to the training of proxy models remains marginal, comprising just 2% of the total computation required for the final training of one 1B model on 25B tokens. We believe this is not just feasible, but also highly efficient, particularly when compared to prior works that necessitate significantly higher computational resources for proxy models.
>
> ---
>
> > W2: The paper only experiments with 1M models with 1B tokens. It is unclear how to decide the size of the proxy model parameter and training token.
> > Q3: How do we decide the size of the proxy model and training tokens?
>
> Thanks for your insightful question! We select a 1M model with 1B tokens as the proxy model, rooted in a strategic methodology for exploring data mixture optimization **with minimal computational resources**. The 1M model represents the minimal proxy model size (which has never been explored by previous works), with just 2 transformer layers, serving as an extremely efficient proxy that allows us to explore diverse data mixtures quickly. By successfully applying RegMix to this ultra-small model, we can gain confidence in the potential scalability of our approach to larger model sizes, providing valuable insights without requiring extensive computational investments.
>
> Recognizing the importance of comprehensive exploration, we also experimented with 60M model proxy models alongside the 1M model, as demonstrated in $\textrm{\color{blue}Figure 13}$ of $\textrm{\color{blue}Appendix F}$. The consistent results of the derived optimized data mixture across these different model scales help validate the stability of our approach. Note that as illustrated in $\textrm{\color{blue}Figure 3}$, increasing the number of training runs can significantly improve the performance of regression models, but not necessarily the tokens. Therefore, we believe using 1M models with 1B tokens proves sufficient for proxy training.
>
> **Back to your question, the decision for the proxy model and training tokens should be a trade-off between computational efficiency and final performance**. Our experiments indicate that, even with a 1000x smaller model and 25x fewer tokens, we can still derive a high-performing data mixture superior to human and previous works. In practice, we recommend first scaling up the number of proxy models (i.e., the number of different data mixtures), instead of increasing the proxy model size and tokens. With the default setting of a 1M model with 1B tokens, we think they should be fine for large-scale experiments. We acknowledge that optimal data mixture may have minor variance between small and large models, but our goal is to establish a practical method that can generate high-performing data mixtures within a feasible computation budget. We will add these explanations to the final version, thanks for your question!

---

> ### Comment · Reviewer_MoX3 · 2024-12-01
>
> Thank you for your detailed responses on the questions.
> I am satisfied with the detailed explanation to address my concerns. I would like to support the work to be accepted.

---

> > ### Author Response · Authors · 2024-12-02
> > **Reply to Reviewer MoX3**
> >
> > Thank you for your constructive review and encouraging words. We really appreciate it! In our final revision, we will polish the paper further to incorporate the valuable insights gained from the rebuttal discussions. Thank you!

---

### Official Review · Reviewer_YiAH · 2024-11-04

**Soundness:** 4
**Presentation:** 4
**Contribution:** 4
**Rating:** 8
**Confidence:** 4

**Summary:**

The work introduces REGMIX, a method for optimizing data mixtures to enhance language model training efficiency. REGMIX treats data mixture selection as a regression task, using small proxy models to predict the performance of different mixtures and identify the best one, enabling larger models to be trained with significantly less compute. Key findings include:
- REGMIX’s mixtures perform as well as or better than those selected by human experts and prior methods like DoReMi, with only a fraction of the compute cost.
- Data mixture has a substantial impact on downstream performance, with single-task performance differences reaching up to 14.6%.
- General web corpora (such as CommonCrawl) outperform traditionally high-quality data like Wikipedia in driving downstream performance.
- Domain interactions are complex and often counterintuitive, highlighting the value of automated approaches like REGMIX.
- Data mixture effects go beyond scaling laws, with REGMIX capturing the complexity by jointly optimizing across all domains.

**Strengths:**

- The paper introduces a novel, regression-based approach for selecting data mixtures that reduces computational costs in language model training. The approach offers an efficient alternative to traditional dynamic or heuristic data allocation methods, making a valuable contribution to the field.

- The paper is technically robust and well-structured, with extensive validation across diverse data scenarios. It empirically supports the rank invariance hypothesis and uses clear, well-structured figures to illustrate the method and results, enhancing reader understanding. REGMIX’s ability to match or outperform other DoReMi and other methods with significantly less compute is a compelling outcome for LLM pre-training efficiency.

- The paper tackles a pertinent problem in LLM pre-training. Given the increasing size of training data and models, this approach could have a significant impact on the field, especially in reducing computational costs and environmental impact.

**Weaknesses:**

- To maximize impact, the authors could highlight specific scenarios where the approach enables previously infeasible experiments due to resource constraints. Also, adding a broader discussion on trade-offs of the method (e.g., scenarios where the rank invariance assumption might not hold) would help readers assess its practical relevance and future applicability.

- The work could have used standardized computation metrics, such as FLOPs or GPU hours, to allow clearer comparison of the method efficiency gains relative to baselines.

**Questions:**

1) Can you further explain why the choice of multiplying the e token distribution by a value from 0.1 to 5.0? Is this a standard practice? Were other ranges tested, and if so, what were the results?
     - Also, could you discuss the rationale behind this range and whether you conducted any sensitivity analyses to determine its impact on the results?

2) Given sufficient computation available, would segmenting domains further (into finer-grained topic-based segments) likely improve model performance or lead to more effective mixture predictions? Do you think that finer segmentation would affect the rank invariance assumption? I suggest the authors to discuss the potential impacts and challenges of a finer-grained domain segmentation in their future work section. This would help address the broader implications and limitations of their approach.

---

> ### Author Response · Authors · 2024-11-21
> **Rebuttal by Authors [1/2]**
>
> > W1: To maximize impact, the authors could highlight specific scenarios where the approach enables previously infeasible experiments due to resource constraints. Also, adding a broader discussion on trade-offs of the method (e.g., scenarios where the rank invariance assumption might not hold) would help readers assess its practical relevance and future applicability.
>
>
> We appreciate your thoughtful feedback on highlighting the practical implications of RegMix. We think a unique contribution of RegMix is its novel use of **ultra-small** proxy models (i.e., 1M parameters) to optimize data mixtures for language model pre-training, an approach previously unexplored in the field. This novel use of ultra-small proxy models reduces computational overhead to less than 2% of final training costs, dramatically lowering barriers for data mixture research within academic budgets. Through the focus on computational efficiency and our commitment to open science (with all datasets and trained models publicly available), we believe RegMix represents a significant advancement toward democratizing research in language model pre-training.
>
> Beyond the methodology contribution, our work delivers several novel empirical insights into data mixture research. We provide the first comprehensive demonstration of significant performance variations across different data mixtures, supported by extensive experiments with 1B-parameter models trained on 64 distinct mixtures and rigorously evaluated across 12 benchmarks. Our results establish the superiority of automatic data mixture optimization over human intuition-based approaches, with PhilPapers serving as an interesting in-depth case study illustrating how domain interactions follow complex patterns that transcend human intuition. We have included these explanations in  $\textrm{\color{blue}Appendix A}$ in the revised submission to maximize the impact of RegMix, and we greatly appreciate your insightful suggestion!
>
> Regarding limitations and future work (detailed in  $\textrm{\color{blue}Appendix B}$), we acknowledge that our investigation of the rank invariance assumption currently focuses on model scales from 1M to 1B parameters. While we aimed to verify the hypothesis at larger scales, establishing statistically meaningful correlations for 3B models would require training 64 different models with 50B tokens each, equivalent to training one 3B model on 3.2T tokens, which significantly exceeds our computational resources.
>
> Following the valuable suggestion of Reviewer `Fgaq`, we are actively conducting experiments with 7B models trained on 100B tokens to compare RegMix against the Human baseline during this rebuttal period. These experiments will provide crucial insights into the scalability of our conclusions to larger models trained on substantially more tokens, and we will share these experimental results as soon as they become available.
>
> Another notable consideration is the token availability challenge. Like most existing data mixing methods, we currently assume unlimited domain data availability. While token availability and data mixture can be controlled in our simulations, we recognize that systematically incorporating real-world data availability constraints presents important theoretical and practical challenges. We see promising directions for future work in exploring data reuse decay coefficients, as introduced in [1], to address limited data scenarios, potentially extending RegMix to broader scenarios.
>
> ---
>
> > W2: The work could have used standardized computation metrics, such as FLOPs or GPU hours, to allow a clearer comparison of the method efficiency gains relative to baselines.
>
> We appreciate your constructive feedback regarding computational metrics. To address your suggestion, we would like to highlight that we have reported the estimated FLOPs in $\textrm{\color{blue}Table 4}$, which provides insights into the computational overhead of different methods. To make it clear, we copy the numbers of DoReMi and RegMix from  $\textrm{\color{blue}Table 4}$ here:
>
> | Method | Average Performance | FLOPs |
> | :---: | :---: | :---: |
> | DoReMi | 46.8 | 3.7e19 |
> | RegMix | 47.3 | 3.5e18 (↓90%) |
>
> As demonstrated above, RegMix achieves a significant 90% reduction in FLOPs compared to the DoReMi baseline, while simultaneously improving average performance. In the revised manuscript, we will further emphasize these computational efficiency gains to provide a more comprehensive evaluation of all approaches.

---

> ### Author Response · Authors · 2024-11-21
> **Rebuttal by Authors [2/2]**
>
> > Q1: Can you further explain why the choice of multiplying the e token distribution by a value from 0.1 to 5.0? Is this a standard practice? Were other ranges tested, and if so, what were the results? Also, could you discuss the rationale behind this range and whether you conducted any sensitivity analyses to determine its impact on the results?
>
> We appreciate your careful reading and thoughtful feedback! The range values of 0.1 to 5.0 are determined more from empirical observation rather than from standard practices, as prior works do not need random data mixtures to fit regression models. These two values are chosen primarily to ensure that the resulting random data mixtures are diverse and meaningful. Although we have not documented the use of other values officially, we conducted experiments with varying ranges previously, and these also showed similarly promising results.
>
> To provide more context on the reason for selecting this particular range, a value leaning towards the lower end of the scale (0.1) creates more skewed data mixtures. Conversely, a value approaching the upper limit (5.0) constructs mixtures more aligned with the original token distributions. This choice accomplishes two principal objectives: **ensuring expansive coverage of the data mixture space** to improve the regression model's generalizability to unseen data mixtures while preserving **effective densities of meaningful information** on each training mixture. Although we do not carry out a formal sensitivity analysis, our empirical findings hint at the regression models' robustness towards variations within this range. Inspired by your suggestion, we plan to conduct a comprehensive study summarizing the impact of these hyper-parameters in our final version.
>
> ---
>
> > Q2: Given sufficient computation available, would segmenting domains further (into finer-grained topic-based segments) likely improve model performance or lead to more effective mixture predictions? Do you think that finer segmentation would affect the rank invariance assumption? I suggest the authors to discuss the potential impacts and challenges of a finer-grained domain segmentation in their future work section. This would help address the broader implications and limitations of their approach.
>
> Thank you for your insightful question! It aligns well with our original intention when designing RegMix – optimally, we aim for it to scale up to more than 100 domains. However, our ambitions are somewhat hampered by the limited availability of a pre-training dataset with a substantial quantity of domains. As a result, our main experiments, as outlined in the main text, are conducted using only 17 domains provided by The Pile dataset.
>
> Nonetheless, inspired by your prompt, we expand our horizons and actively conduct a more comprehensive examination of RegMix's scalability. During the rebuttal period, we do a preliminary study covering 100 domains. The selection of these domains is based on their respective base URL domains and the availability of tokens. This effort was specifically intended to highlight the adaptability of RegMix across an extended range of domains, up to as many as 100.
>
> In more detail, we analyze the token availability across different URL domains using the most recent pre-training dataset, FineWeb [2]. Each URL domain is considered a separate domain, and we carry out data mixtures on these domains. We provide a sample of these domains below:
>
> ```
> articles.latimes.com
> blogs.wsj.com
> en.wikipedia.org
> everything2.com
> ideas.repec.org
> latimesblogs.latimes.com
> news.bbc.co.uk
> ...
> ```
> To evaluate the effectiveness of RegMix, we train 1,000 proxy models of 1M parameters, fit them with regression models, and examine the rank correlation between the rankings predicted by the regression model and the actual rankings of 64 unseen data mixtures on these domains. Taking the validation loss on the `en.wikipedia.org` domain as a demonstration, and following our main experiments validating on both 1M and 60M models, RegMix delivers the subsequent regression performance across these 100 domains:
>
> || 1M ($\rho$ $\uparrow$)|1M (MSE $\downarrow$)|60M ($\rho$ $\uparrow$)|
> |----|----|----|----|
> |Linear| 90.33| 0.12| 88.64|
> |LightGBM| 99.53| 0.02| 98.80|
>
> As shown above, consistent with our findings on The Pile dataset (17 domains), RegMix demonstrates a commendable performance even when scaled up to 100 domains. With LightGBM regression, it continues to achieve a high correlation on unseen data mixtures for both 1M models (correlation of 99.53%) and 60M models (correlation of 98.80%). These results indicate the applicability of RegMix over a wide array of domains.
>
> We have also updated the table above along with a visualization in  $\textrm{\color{blue}Appendix H}$ in the revised submission.
>
> [1] Muennighoff et al., "Scaling Data-Constraint Language Models", NeurIPS 2023.
>
> [2] Penedo et al, "The FineWeb Datasets: Decanting the Web for the Finest Text Data at Scale", 2024.

---

> > ### Comment · Reviewer_YiAH · 2024-12-02
> > **Reply to authors**
> >
> > I acknowledge the authors response, I thank them for the efforts in addressing the issues pointed out. I also acknowledge the authors response to other reviewers, which addressed other relevant issues. I believe the modifications made contribute to make the paper even more stronger. I keep my score, I believe the paper should be accepted at the main conference.

---

> > > ### Author Response · Authors · 2024-12-02
> > > **Reply to Reviewer YiAH**
> > >
> > > Thank you for your thoughtful and constructive review! We deeply appreciate the valuable insights and feedback provided during the rebuttal discussions. In our final revision, we will carefully refine the paper, ensuring that we fully incorporate the suggestions and address the key points raised. Thank you again!

---

### Author Response · Authors · 2024-11-21
**Rebuttal to All Reviewers**

We thank all reviewers for their constructive feedback, and we have learned a lot from the feedback and have responded to each reviewer individually. We are very excited to receive this positive feedback from all of you. Here we want to highlight some new experimental results that may interest all reviewers.

- **New Experiments on 100 Domains**: We have extended RegMix to 100 domains, achieving an impressive correlation of 98.80% between the predicted ranks and true ranks of data mixtures. This demonstrates the scalability of RegMix when applied to a larger number of domains.

- **New Experiments with 280M Models**: We have added new experiments with 280M models and systematically demonstrate the rank invariance hypothesis across four model scales (1M, 60M, 280M, 1B) and three token scales (1B tokens, 5B tokens, 25B tokens).

- **Ongoing Experiment - 7B Model with Optimized Data Mixture**: We are currently applying the optimized data mixture from RegMix to a 7B model trained on 100B tokens. Once the results are available, we will compare its performance against the human baseline.

- **Ongoing Experiment - 1B Level Proxy Models**: We are also investigating using 1B models as proxy models to generate optimized data mixtures for training 7B models. This experiment aims to provide additional insights into the selection of appropriate proxy model sizes. We will report back once the results are available.

- **Paper Revision**: We have made the following updates:
   - $\textrm{\color{blue}Appendix A}$: describe the broader impact of our paper.
   - $\textrm{\color{blue}Appendix B}$: include a detailed discussion on the limitations around the rank invariance hypothesis.
   - $\textrm{\color{blue}Appendix H}$: extend RegMix to 100 domains.
   - $\textrm{\color{blue}Appendix I}$: add the pseudocode for RegMix.
   - $\textrm{\color{blue}Appendix J}$: provide additional empirical evidence supporting the rank invariance hypothesis.

---

### Author Response · Authors · 2024-11-25
**More Experimental Results by Authors**

Hi all reviewers, thank you once again for your valuable comments and suggestions, which have been very helpful for us! We understand that this is a particularly busy time, and we greatly appreciate it if you could take a moment to review our additional experiments and response and let us know if they adequately address your concerns. Should there be any additional feedback, we will do our best to incorporate it in the next few days.

As previously committed, we have completed the experiments and are now reporting our findings:
- $\textrm{\color{blue}Appendix K}$: Comparing RegMix and Human on a 7B model over 100B tokens, RegMix continues to outperform Human by a significant margin, with the ability to accelerate performance by up to 75% on some benchmarks.
- $\textrm{\color{blue}Appendix L}$: Investigating whether it could bring more gains by training large proxy models (e.g., 1B). The conclusion is that our 1M proxy model is already strong enough, so we recommend researchers to begin with the ultra small models for data mixture optimization.

## Appendix K: Scaling RegMix to 7B Models over 100B Tokens

In order to further validate the effectiveness of our method on larger models, we conducted an experiment using Human baseline and RegMix data mixtures on a 7B model trained on 100B tokens. The results, summarized in ${\textrm{\color{blue}Table 10}}$  (see below), demonstrate that RegMix still outperforms the Human baseline, achieving an average performance boost of 2%:

| Benchmark | Human | Our Method |
| --- | --- | --- |
| Social IQA | 41.6 | 43.3 |
| HellaSwag | 55.0 | 63.3 |
| PiQA | 72.4 | 75.3 |
| OpenBookQA | 34.1 | 36.7 |
| Lambada | 44.9 | 51.0 |
| SciQ | 91.7 | 91.2 |
| ARC Easy | 63.5 | 65.4 |
| COPA | 75.0 | 80.1 |
| RACE | 35.1 | 36.6 |
| LogiQA | 25.7 | 24.1 |
| QQP | 58.9 | 56.1 |
| WinoGrande | 58.6 | 60.7 |
| MultiRC | 52.0 | 51.0 |
| **Average** | **54.5** | **56.5 (+2.0)** |

To provide a more detailed illustration, we benchmarked the downstream performance of RegMix and Human on every dataset at intervals of 25B tokens in $\textrm{\color{blue}Figure 16}$ of $\textrm{\color{blue}Appendix K}$. The results show that RegMix can significantly speed up pre-training, with a **50% acceleration on most benchmarks (e.g., HellaSwag)** and **up to 75% on some benchmarks (e.g., PiQA)**. Notably, the performance boost does not decrease with the amount of training tokens. However, we also observed that both RegMix and Human struggle to improve on certain benchmarks (e.g., MultiRC), even with increased token amounts. These findings suggest that RegMix mainly benefits downstream tasks whose performance increases with the amount of training data, but may not improve tasks that do not follow scaling laws. This observation is intriguing and worthy for further investigation.

---

## Appendix L: Using Larger Proxy Model

We conducted a preliminary study on the impact of proxy model size on effectiveness. Specifically, we compared two configurations: (1) 128 proxy models of 1B parameters each, and (2) 512 proxy models of 1M parameters each (the setting used in our main experiments). Both configurations used 1B training tokens per proxy model. We limited our investigation to these configurations due to computational constraints that prevented us from exploring scenarios with more 1B-parameter proxy models.

To evaluate these configurations, we used their respective optimized data mixtures to train two 7B models on 100B tokens and compared their performance. The results, summarized in $\textrm{\color{blue}Table 11}$ (see below), show that both proxy settings achieved similar average performance across downstream tasks. This suggests that increasing proxy model size, even with fewer proxy models, can maintain competitive performance. However, given that the 1B proxy models do not significantly outperform the 1M proxy models, and considering that they incur over much more computational overhead, we recommend prioritizing a larger number of smaller proxy models over fewer larger ones. Based on our findings, we suggest practitioners begin with ultra-small proxy models (e.g., 1M parameters in our setting) as a starting point to optimize data mixtures for language model pre-training.

| **Benchmark** | **RegMix** (1B as proxy) | **RegMix** (1M as proxy) |
|--------------|------------------------------|------------------------------|
| Social IQA | 43.4 | 43.3 |
| HellaSwag | 62.9 | 63.3 |
| PiQA | 75.1 | 75.3 |
| OpenBookQA | 36.2 | 36.7 |
| Lambada | 50.0 | 51.0 |
| SciQ | 91.2 | 91.2 |
| ARC Easy | 65.9 | 65.4 |
| COPA | 79.6 | 80.1 |
| RACE | 35.6 | 36.6 |
| LogiQA | 23.9 | 24.1 |
| QQP | 56.6 | 56.1 |
| WinoGrande | 60.7 | 60.7 |
| MultiRC | 51.6 | 51.0 |
| **Average** | **56.4** | **56.5** |

---

### Author Response · Authors · 2024-11-27
**Comment from Authors**

Dear Reviewers,

Since the rebuttal period has been extended to December 2, we are excited to have another week to  welcome any additional questions or clarifications you may have regarding our submission. Based on the previous round of reviews, we have made significant improvements to RegMix:

- Expanding the experiments to 1,000 domains for RegMix and verify its effectiveness
- Verifying performance improvement on 7B-level models over 100B tokens
- Systematically verifying the rank invariance hypothesis on model scales and more token counts
- Investigating proxy model sizes at the 1B level with new insights

We eagerly welcome the opportunity to further discuss the potential of RegMix. If you have any specific inquiries or require additional information, please do not hesitate to share them. We are sincerely grateful to all reviewers for their constructive, helpful, and timely feedback.

Best Regards,

Authors

---

### Meta-Review · Area_Chair_GMhU · 2024-12-16

**Metareview:**

This paper introduces REGMIX, a novel method for optimizing data mixtures to improve the efficiency of pre-training LLMs. The method frames data mixture selection as a regression task, using small proxy models trained on diverse data mixtures to predict the best-performing mixture for larger-scale training. REGMIX presents a significant advance in automating data mixture selection, reducing computational costs while achieving or surpassing the performance of prior methods and human expertise. Detailed key findings and contributions include:
- Methodology: REGMIX uses small models to evaluate various data mixtures, fits a regression model to predict their performance, and identifies the optimal mixture for larger-scale training.
- Efficiency and Effectiveness (showed empirically): REGMIX-selected mixtures perform as well as or better than mixtures chosen by human experts or prior methods like DoReMi while using only 10% of the compute budget. The method also demonstrates improved downstream task performance, with single-task performance differences of up to 14%. These findings were obtained with comprehensive experiments, including testing with a 1B parameter model, and comparing two regression methods for modeling data mixtures.
- Key Insights: Data composition has a substantial impact on LLM performance, often in counterintuitive ways, emphasizing the importance of automated approaches like REGMIX. Authors also found general web corpora (e.g., CommonCrawl) outperform traditionally high-quality datasets like Wikipedia for downstream performance, and REGMIX can capture complex interactions across data domains, revealing insights that go beyond existing scaling laws.


Strength of this paper

- Novel and Effective Approach: The paper introduces a novel regression-based method, REGMIX, for optimizing data mixtures in LLM pre-training, reducing computational costs while maintaining or surpassing performance compared to methods like DoReMi and human selection. By framing data mixture selection as a regression task, the method avoids complex optimization or heuristic approaches, making it computationally efficient and scalable.
- Experimental Rigor: Extensive experiments to cover diverse data scenarios and validate the method, including training 512 small proxy models on diverse data mixtures and testing with a 1B parameter model trained on 25B tokens. The authors empirically support the rank invariance hypothesis, demonstrating consistency in mixture rankings between small and large models.
- Scalability and Practicality: The method enables parallel training of small proxy models, significantly reducing the time and compute required compared to traditional approaches. It is simple to implement and efficient, making it highly practical for real-world applications.
- Impact and Relevance: The approach addresses a critical problem in LLM pre-training, offering a solution that reduces computational and environmental costs, which is increasingly important as models and datasets grow larger.
- The paper is well-written, clearly explaining the methodology, experiments, and findings.

Weakness of this paper

Several reviewers raised few concerns/limitations of this paper. By addressing these limitations, the paper could strengthen its experiment and expand impact.

Weaknesses of the Paper:
- Experimental scale and settings: The experiments are limited to models with 1M parameters trained on 1B tokens and validated with 1B parameter models trained on 25B tokens. The scalability to much larger datasets or model sizes is untested. It is also unclear how to determine the optimal size of the proxy model parameters or the training tokens, which limits the reproducibility and generalization of the method. The paper does not use standardized computation metrics like FLOPs or GPU hours, making it harder to quantify efficiency gains relative to baselines.
- Impact, applicability, and reproducibility: The authors could better emphasize specific scenarios where the approach enables previously infeasible experiments due to resource constraints, or discuss on trade-offs, such as when the rank invariance assumption may not hold, to help readers better understand the method's limitations and future applicability. Key details about the regression model, such as the number of data points used for fitting, are not clearly explained. Adding pseudocode or open-sourced scripts would improve clarity and reproducibility.
- Baseline Comparisons: The mixture weights for DoReMi were directly taken from the original paper, but differences in model or data processing could affect the results. Re-learning these weights in the current setup would provide a fairer comparison.

**Additional Comments On Reviewer Discussion:**

Above summarized the strength and weaknesses raised by reviewers. Most of the weaknesses were addressed via further discussion and more experiment results. Given the relatively positive ratings, the strengthens summarized above, and mitigated concern on weaknesses, I recommend to accept this paper.

---

### Decision · Program_Chairs · 2025-01-22

Accept (Spotlight)